# FUNDAMENTAL BOUNDS ON EFFICIENCY-CONFIDENCE TRADE-OFF FOR TRANSDUCTIVE CONFORMAL PREDICTION

## ABSTRACT

Transductive conformal prediction addresses the simultaneous prediction for multiple data points. Given a desired confidence level, the objective is to construct a prediction set that includes the true outcomes with the prescribed confidence. We demonstrate a fundamental trade-off between confidence and efficiency in transductive methods, where efficiency is measured by the size of the prediction sets. Specifically, we derive a strict finite-sample bound showing that any non-trivial confidence level leads to exponential growth in prediction set size for data with inherent uncertainty. The exponent scales linearly with the number of samples and is proportional to the conditional entropy of the data. Additionally, the bound includes a second-order term, dispersion, defined as the variance of the log conditional probability distribution. We show that this bound is achievable in an idealized setting. Finally, we examine a special case of transductive prediction where all test data points share the same label. We show that this scenario reduces to the hypothesis testing problem with empirically observed statistics and provide an asymptotically optimal confidence predictor, along with an analysis of the error exponent.

## 1 INTRODUCTION

Modern decision systems often need to *predict many outcomes at once* and act on the *joint result*. Examples include certifying components in a quality-control batch, screening biological samples for pathogens, or approving software changes before release. In such settings, the cost of even a single error can be high, making *distribution-free guarantees* on the *entire vector of predictions* essential.

Conformal prediction (CP) (Vovk et al., 2022) offers a principled framework for constructing prediction sets with finite-sample, distribution-free coverage guarantees under minimal assumptions. Typically, CP methods operate on individual input–output pairs, where each input $X$ is associated with a label $Y$. However, many real-world systems require *joint guarantees* across multiple predictions, motivating the study of *transductive conformal prediction (TCP)*. TCP constructs a joint prediction set for a batch of test inputs $X_1, \ldots, X_n$, ensuring that the corresponding label vector $Y_1, \ldots, Y_n$ lies within the set with a prescribed confidence level (e.g., 95%).

While TCP offers stronger guarantees, it raises a fundamental question: how small can such joint sets be, on average, while still guaranteeing coverage? This question is not merely practical, it probes the limits of uncertainty quantification in multi-output prediction. Our paper addresses this challenge and makes the following contributions:

- **Fundamental lower bound:** We prove that for any non-trivial confidence level, the expected size of any valid joint prediction set must grow *exponentially* with the number of test points. The growth rate is governed by the conditional entropy $H(Y|X)$ and a second-order term we call *dispersion*, which captures the variance of the log-conditional probabilities.

- **Achievability:** We show that this bound is *tight* by constructing an idealized predictor (with oracle access to $P(Y|X)$) that matches the first and second-order terms.

- **Homogeneous-label setting:** When all test points share the same unknown label, a scenario relevant to safety-critical applications, the problem reduces to *hypothesis testing with*

*empirically observed statistics.* We derive an *asymptotically optimal confidence predictor* based on thresholding a generalized Jensen–Shannon divergence and characterize its error exponent.

These results hold under minimal assumptions: they apply to any conformity score, extend to a larger class of efficiency metrics beyond prediction set size, and are validated by experiments showing their relevance in finite-sample regimes, highlighting inefficiencies in existing transductive methods.

## 2   TRANSDUCTIVE CONFORMAL PREDICTORS AND HYPOTHESIS TESTING

To prepare for our main results, this section contrasts standard conformal prediction (CP), which offers marginal coverage for individual predictions, with its transductive extension (TCP) that provides joint guarantees across a batch. We begin by introducing the necessary notation.

**Notation.**   In this paper, the random variables are denoted by capital letters $X_1, X_2, ...$ and their realization by $x_1, x_2, \ldots$, and the vectors and matrices are denoted by bold letters as $\boldsymbol{X}, \boldsymbol{x}$. $X_i^j$ denotes the tuple $(X_i, \ldots, X_j)$. We use $P(Y|X)$ to denote the conditional distribution of labels $Y$ given samples $X$. We use $P$ as well to denote the distribution over $X$ and $Y$. The logarithms are all assumed to be natural logarithms, unless otherwise stated. The entropy $H(X)$ is defined as $\mathbb{E}[-\log P(X)]$, the conditional entropy defined as $H(Y|X) := \mathbb{E}[-\log P(Y|X)]$. The Kullback-Leibler divergence is defined as $D(Q\|P) := \mathbb{E}_Q[\log dQ/dP]$. $Q(\cdot)$ is the Gaussian Q-function defined as $Q(t) := \mathbb{P}(X > t)$ for $X$ a standard normal distribution.

**From Standard to Transductive Conformal Prediction (TCP).**   Standard conformal prediction (CP) constructs a *per-input* prediction set that contains the true label with probability at least $1 - \alpha$, under exchangeability. Formally, consider a sequence of labeled examples $Z_1^m = ((X_i, Y_i) : i \in [m])$, where $X_i \in \mathcal{X}, Y_i \in \mathcal{Y}$ with $M$ distinct classes, i.e., $|\mathcal{Y}| = M$, and a test sample $X_{m+1}$ with unknown true label $Y_{m+1}$. Standard CP produces a set $\Gamma^\alpha(X_{m+1})$ that satisfy *marginal coverage*: $\mathbb{P}(Y_{m+1} \notin \Gamma^\alpha(X_{m+1})) \leq \alpha$. These sets are obtained by thresholding p-values: for each input, all labels with p-value above $\alpha$ are included. A popular variant, split CP (Papadopoulos et al., 2002; Lei et al., 2018), computes these p-values using pretrained predictor and a separate calibration set. More generally, CP can be formalized through *transducers*, which map sequences of labeled examples to p-values in $[0, 1]$, providing a unified view of conformal methods.

While the *marginal* guarantee of standard CP is often sufficient for isolated decisions, many applications require system-level guarantees, such as maintaining a global missed-detection constraint in autonomous driving or ensuring consistency in ranking tasks (Fermanian et al., 2025). In these settings, a single error can invalidate the entire outcome, motivating *joint* guarantees. Transductive conformal prediction (TCP) addresses this by constructing a joint prediction set for the whole test batch (Vovk, 2013; Vovk et al., 2022). Given $Z_1^m$ and a batch of test samples $X_{m+1}^{m+n} = (X_{m+1}, \ldots, X_{m+n})$, a (transductive) confidence predictor outputs a set of candidate label vectors $(\boldsymbol{Y}_{m+1}, \ldots, \boldsymbol{Y}_{m+n})$ such that the error probability of the predictor $P_e$ is bounded

$$P_e = \mathbb{P}\left(Y_{m+1}^{m+n} \notin \Gamma^\alpha(Z_1^m, X_{m+1}^{m+n})\right) \leq \alpha. \tag{1}$$

If the predictor satisfies the significance level $\alpha$, we say it has confidence $1 - \alpha$. Some works instead use the *False Coverage Proportion (FCP)* as the error (Fermanian et al., 2025), which measures the average per-sample error rather than the joint error over the entire test set. This is a more relaxed criterion than the one adopted here and in (Vovk et al., 2022) (see Appendix G for details).

Operationally, TCP extends the conformal principle from single examples to sequences: instead of computing p-values for individual labels, we compute them for entire candidate label sequences using *transductive conformity scores*. These scores assess how well a proposed joint labeling fits the observed data and the test batch. Thresholding these p-values yields a joint confidence set that guarantees coverage for all test points simultaneously. A common baseline is to aggregate per-sample p-values via Bonferroni: build $\Gamma^{\alpha/n}(X_{m+i})$ for each test point and take the Cartesian product $\prod_{i=1}^n \Gamma^{\alpha/n}(X_{m+i})$, which satisfies eq. 1 but can be inefficient as $n$ grows (cf. our experiments).

**Efficiency vs. Confidence.**   While eq. 1 guarantees *confidence*, practitioners also care about *efficiency*: how large the joint prediction set is on average. These two objectives are inherently in

tension, and understanding this trade-off is among the main objectives of this work. To that end, we measure efficiency by the cardinality of $\Gamma^\alpha(Z_1^m, X_{m+1}^{m+n})$, though our results extend to other notions of efficiency (see Appendix C). Of particular interest is the *efficiency rate*, which captures the exponential growth of the expected prediction set size as the number of test samples $n$ increases.

**Definition 2.1.** The *efficiency rates* of a transductive conformal predictor are

$$\gamma_{n,m} := \frac{1}{n} \log \mathbb{E} \big| \Gamma^\alpha(Z_1^m, X_{m+1}^{m+n}) \big|, \qquad \gamma_m^+ := \limsup_{n \to \infty} \gamma_{n,m}, \ \gamma_m^- := \liminf_{n \to \infty} \gamma_{n,m}.$$

If these limits coincide, we denote the common value by $\gamma_m$.

**Research Questions and Road Map.** Building on the formalism above, we revisit the interpretation in Correia et al. (2024), where split CP is viewed through the lens of list decoding (Wozencraft, 1958). In this setting, the model output is treated as a noisy observation of the ground truth label, which enabled the authors to establish information-theoretic inequalities linking the efficiency of conformal prediction, as measured by the expected size of the prediction set, to the conditional entropy $H(Y|X)$ of the labeling distribution.

However, two fundamental questions remain open. First, the **efficiency-confidence trade-off** characterized in Correia et al. (2024) in terms of $H(Y|X)$ and the logarithm of the expected prediction set size was not tight. This raises the question: *Can we derive tighter bounds on the efficiency-confidence trade-off, and characterize conditions under which these bounds are achievable?*

Second, split CP's reliance on a separate calibration set underutilizes the available data, as training samples are discarded for calibration. This motivates a broader question: *What is the information-theoretically optimal way to construct confidence predictors that leverage the entire dataset without sacrificing validity?* Addressing this question requires bridging inductive and transductive paradigms and exploring connections with hypothesis testing under empirically observed statistics.

In the rest of this paper, we tackle these challenges in two steps. First, we establish fundamental bounds on the efficiency-confidence trade-off in the general transductive setting, capturing both asymptotic and finite-sample regimes and revealing a phase transition governed by the conditional entropy. Next, we consider a structured scenario where all test samples share the same label, reducing the problem to multiple-hypothesis testing. This enables us to design an asymptotically optimal confidence predictor based on generalized Jensen-Shannon divergence, shedding light on the interplay between confidence control and efficiency in practice.

## 3 FUNDAMENTAL BOUNDS ON EFFICIENCY-CONFIDENCE TRADE-OFF

In Correia et al. (2024), the authors derived new information-theoretic bounds that connected conformal prediction to list decoding. The bounds involved terms related to the efficiency of conformal prediction and the conditional entropy or KL-divergence terms and leveraged Fano's inequality and data processing inequality. In this section, we derive new bounds that can generally lead to tighter bounds. The proofs are all relegated to Appendix B.

Consider the case where the error probability $P_e$ is not exceeding $\alpha$, that is $P\left(Y_{m+1}^{m+n} \notin \Gamma^\alpha(Z_1^m, X_{m+1}^{m+n})\right) \leq \alpha$. We have the following result.

**Theorem 3.1.** *Consider a transductive conformal predictor $\Gamma^\alpha(Z_1^m, X_{m+1}^{m+n})$ given a labeled dataset $Z_1^m$ and test samples $X_{m+1}^{m+n}$ with unknown labels $Y_{m+1}^{m+n}$. If the predictor has the confidence $1 - \alpha$, then for any $\beta \in (0, 1)$, we have:*

$$\mathbb{P}(P(Y_{m+1}^{m+n}|X_{m+1}^{m+n}) \leq \beta) \leq \alpha + \beta \mathbb{E}(|\Gamma^\alpha(Z_1^m, X_{m+1}^{m+n})|) \tag{2}$$

We have focused on the prediction set size as the notion of efficiency. It is possible to generalize this to any measure of efficiency on the prediction set. We show these results in Appendix C.

The proofs are all presented in the Appendix. The original theorem follows from this one using a counting measure. These theorems can be used to derive bounds for the efficiency-confidence trade-off for transductive conformal prediction. We start with the following theorem.

**Theorem 3.2.** *Consider a transductive conformal predictor $\Gamma^\alpha(Z_1^m, X_{m+1}^{m+n})$ with confidence level $1 - \alpha_n$ for $n$ test samples. Then, we have:*

1. *If the asymptotic confidence is non-trivial, i.e., $\liminf_{n\to\infty}(1-\alpha_n) > 0$, the efficiency rate satisfies:*

$$\gamma_m^- \geq H(Y|X).$$

2. *If $\gamma_m^- < H(Y|X)$, then the confidence vanishes asymptotically to zero:*

$$\lim_{n\to\infty}(1-\alpha_n) = 0.$$

*Remark* 3.3. The theorem states a fundamental asymptotic trade-off between the confidence level and the prediction set size. Roughly, the prediction set size needs to grow exponentially at least as $e^{nH(Y|X)}$ to avoid a non-trivial confidence level. Another insight is that asymptotically, there is a phase transition at the efficiency rate $H(Y|X)$ below which it is impossible to get non-trivial confidence. The result does not indicate anything regarding the impact of the asymptotic confidence level on the set size. Indeed, it can be seen that $\liminf_{n\to\infty}\frac{1}{n}\log(1-\alpha_n) = 0$ for any non-trivial $\alpha_n$. In other words, it seems that it suffices to have $\gamma_m^- \geq H(Y|X)$ to get any non-trivial confidence *asymptotically*. In the case of classical conformal prediction, where the prediction set of each sample is predicted independently, the result means that the expected prediction set size is greater than or equal to $e^{H(Y|X)}$. A similar observation was reported in Correia et al. (2024).

**Non-asymptotic results.** Similar to the analysis of finite block length in Polyanskiy et al. (2010), we can derive a non-asymptotic bound for the efficiency-confidence trade-off using the growth rate of the average prediction set size.

**Theorem 3.4.** *For a transductive conformal predictor with the confidence level $1-\alpha$, consider the efficiency rate defined as:*

$$\gamma_{n,m} := \frac{1}{n}\log\mathbb{E}|\Gamma^\alpha(Z_1^m, X_{m+1}^{m+n})|,$$

*which is the growth exponent of the prediction set size. Then for any $n$, we have:*

$$\log\Delta + nH(Y|X) + \sqrt{n}\sigma Q^{-1}\left(\alpha + \frac{\rho}{\sqrt{n}\sigma^3} + \Delta\right) \leq n\gamma_{n,m}$$

*if $\alpha + \frac{\rho}{\sqrt{n}\sigma^3} + \Delta \in [0,1]$ where $\Delta > 0$ and $Q(\cdot)$ is the Gaussian Q-function, and:*

$$\sigma := \left(\mathrm{Var}\left(\log P(Y|X)\right)\right)^{1/2} = \left(\mathbb{E}\left(\log P(Y|X) + H(Y|X)\right)^2\right)^{1/2} \tag{3}$$

$$\rho := \mathbb{E}\left(\left|\log P(Y|X) + H(Y|X)\right|^3\right). \tag{4}$$

The non-asymptotic results leverage the Berry-Esseen central limit theorem to characterize the sum $\sum_{i=1}^n \log\mathbb{P}(Y_{m+i}|X_{m+i})$ in Theorem 3.1. We call the term $\sigma$, *the dispersion* following a similar name used in finite block length analysis of information theory (Polyanskiy et al., 2010; Strassen, 1962). Note that these bounds do not assume anything about the underlying predictor, and therefore do not show any dependence on the number of training samples $m$. The underlying method might as well have access to the underlying distributions $P(Y|X)$.

To use the above bound, we provide an approximation by ignoring some constant terms that diminish with larger $n$. The approximate bound can be easily computed and is given as follows

$$\boxed{n\gamma_{n,m} \geq nH(Y|X) + \sqrt{n}\sigma Q^{-1}(\alpha) - \frac{\log n}{2} + O(1).} \tag{5}$$

We provide the derivation details in the Appendix.

**On achievability of the bounds.** In this part, we argue that the provided bound are achievable. Suppose that we know the underlying probability distribution $P(Y|X)$, which corresponds to the idealized setting in Vovk et al. (2022). Upon receiving test samples $X_1, \ldots, X_n$, we can construct the confidence sets as follows:

$$\Gamma^\alpha(X_1^n) := \left\{(y_1, \ldots, y_n) : \prod_{i=1}^n P(y_i|X_i) \geq \beta\right\}.$$

Similarly, the efficiency rate is defined as $\gamma_n := \frac{1}{n}\log\mathbb{E}[|\Gamma^\alpha(X_1^n)|]$. We show that with proper choice of $\beta$ we can achieve the lower bound, ignoring the logarithmic terms, at a given significance level $\alpha$. The definition of $\rho$ and the dispersion $\sigma$ is similar to Theorem 3.4.

**Theorem 3.5.** *For the confidence set* $\Gamma^\alpha(X_1^n)$ *defined above, and for* $\alpha \geq \rho/\sqrt{n}\sigma^3$, *there is a choice of* $\beta$ *that achieves the confidence* $1 - \alpha$ *at the efficiency rate* $\gamma_n$ *satisfying:*

$$n\gamma_n \leq nH(Y|X) + \sqrt{n}\sigma Q^{-1}(\alpha) + O(1).$$

As it can be seen, knowing the underlying probability distribution, the prediction set size can be bounded, and therefore, the achievability bound matches the first and second order term in the converse bound. For the proof and more details see Section D.

## 4 Hypothesis Testing with Empirically Observed Statistics

In the previous section, we studied the case where multiple test samples could have different labels. In safety-critical applications, however, a single prediction task is often repeated across multiple samples from the same experiment to enhance robustness. In such cases, it is reasonable to assume that all test samples share the same label. Formally, we assume a balanced training dataset with $M$ classes and $N$ samples per class, denoted as $\mathbf{X}_i = (X_{i,1}, \ldots, X_{i,N})$ for each $i \in [M]$, resulting in a total training set size of $m = N \cdot M$. At test time, we receive $n$ samples $\mathbf{X}_{\text{test}} = (X_{m+1}, \ldots, X_{m+n})$, all from a single, unknown class. The goal is to determine the label of these test samples.

Assume that samples $X_{i,j}$ from class $j$ follow a distinct distribution over $\mathcal{X}$, denoted by $P_i$ for $i = 1, 2, \ldots, M$. These distributions are unknown; we only have access to their samples via the training data. The transductive prediction problem, in this context, reduces to identifying which class in the training data shares the same distribution as the test samples. This is equivalent to a multiple hypothesis testing problem with empirically observed statistics (Ziv, 1988; Gutman, 1989; Zhou et al., 2020), where each hypothesis corresponds to a class label. We consider the hypotheses $H_i$ for $i = 1, 2, \ldots, M$ where the test sequence $\mathbf{X}_{\text{test}}$ is generated according to the distribution $P_i$, i.e., the same distribution used to generate $X_i^N$. In the context of transductive conformal prediction, the confidence predictor returns a list of hypotheses. This setup introduces two simplifications compared to general transductive learning: (1) the training set is balanced, with the same number of samples $N$ per class, and (2) all test samples $\mathbf{X}_{\text{test}}$ are assumed to originate from the same distribution.

**Binary Classification without Confidence - Asymptotic Results.** We first review the classical results. For the rest, we assume $N = \alpha \cdot n$. In binary classification, the decision rule is given by the mapping $\psi_n : \mathcal{X}^{2 \times N} \times \mathcal{X}^n \to \{H_1, H_2\}$. The decision rule partitions the space into 2 disjoint regions without reporting confidence. Two errors corresponding to false alarm (type I) and missed detection (type II) arise in hypothesis testing, given by:

$$\beta_1(\psi_n|P_1, P_2) := \mathbb{P}_{P_1}(\psi_n(\mathbf{X}_1, \mathbf{X}_2, \mathbf{X}_{\text{test}}) = H_2), \tag{6}$$

$$\beta_2(\psi_n|P_1, P_2) := \mathbb{P}_{P_2}(\psi_n(\mathbf{X}_1, \mathbf{X}_2, \mathbf{X}_{\text{test}}) = H_1), \tag{7}$$

where $\mathbb{P}_{P_i}$ means that the test sequence follows the distribution $P_i$ for $i = 1, 2$. The optimal decision rule for this problem has been studied in the literature. To state the main result, we will introduce the following quantity, known as the generalized Jensen-Shannon divergence:

$$\text{GJS}(P_i, P_j, \alpha) = \alpha D\left(P_i \middle\| \frac{\alpha P_i + P_j}{1 + \alpha}\right) + D\left(P_j \middle\| \frac{\alpha P_i + P_j}{1 + \alpha}\right). \tag{8}$$

For this problem, Gutman suggested the following test in Gutman (1989):

$$\psi_n^{\text{Gutman}}(\mathbf{X}_1, \mathbf{X}_2, \mathbf{X}_{\text{test}}) = \begin{cases} H_1 & \text{if } \text{GJS}(T_{\mathbf{X}_1}, T_{\mathbf{X}_{\text{test}}}, \alpha) \leq \lambda \\ H_2 & \text{if } \text{GJS}(T_{\mathbf{X}_1}, T_{\mathbf{X}_{\text{test}}}, \alpha) > \lambda \end{cases} \tag{9}$$

where $T_{\mathbf{X}}$ is the type of the sequence $\mathbf{X}$, i.e., its empirical probability mass function. See eq. A for more details. Note that the generalized Jensen-Shannon divergence for types gets the following form:

$$\text{GJS}(T_{\mathbf{X}_1}, T_{\mathbf{X}_{\text{test}}}, \alpha) = D(T_{\mathbf{X}_{\text{test}}} \| T_{\mathbf{X}_1, \mathbf{x}_{\text{test}}}) + \alpha D(T_{\mathbf{X}_1} \| T_{\mathbf{X}_1, \mathbf{x}_{\text{test}}}).$$

This test is known to be asymptotically optimal in the following sense. First, for any distributions $P_1$ and $P_2$, we have:

$$\liminf_{n \to \infty} -\frac{1}{n} \log \beta_1(\psi_n^{\text{Gutman}}|P_1, P_2) \geq \lambda \tag{10}$$

$$\liminf_{n \to \infty} -\frac{1}{n} \log \beta_2(\psi_n^{\text{Gutman}}|P_1, P_2) = F(P_1, P_2, \alpha, \lambda), \tag{11}$$

where

$$F(P_1, P_2, \alpha, \lambda) := \min_{\substack{(Q_1, Q_2) \in \mathcal{P}(\mathcal{X})^2 \\ \text{GJS}(Q_1, Q_2, \alpha) \leq \lambda}} D(Q_2 \| P_2) + \alpha D(Q_1 \| P_1). \tag{12}$$

Next, consider any other decision rule $\phi_n$ that *uniformly* controls the error exponent of $\beta_1(\psi_n | P_1, P_2)$ similar to Gutman, namely

$$\forall (P_1, P_2) \in \mathcal{P}(\mathcal{X})^2 : \liminf_{n \to \infty} -\frac{1}{n} \log \beta_1(\psi_n | P_1, P_2) \geq \lambda.$$

Then, the second error is always worse than Gutman's test: $\beta_2(\psi_n | P_1, P_2) \geq \beta_2(\psi_n^{\text{Gutman}} | P_1, P_2)$. The proof can be found in Dembo & Zeitouni (2009, Theorem 2.1.10) and is based on Sanov's theorem. The first conclusion of this result is that with Gutman's test, we will asymptotically have prediction sets with a single true label in the set, as both probability of errors vanishes. The result in this sense is not surprising. Asymptotically, we have sufficient samples at both training and test times ($n, N \to \infty$) to estimate the distributions precisely. Note that the prediction set always has the cardinality of one, so there is no confidence associated with it. Also, asymptotically, the probability error decreases exponentially, which means that the set of size one is asymptotically achievable with an arbitrary level of confidence if $F(P_1, P_2, \alpha, \lambda) \neq 0$. See Zhou et al. (2020) for non-asymptotic results and further discussions.

By controlling $\lambda$, we can maintain the decay rate of the error of the first type; however, this comes at the cost of a worse error rate for the error of the second type. This term would dominate the Bayesian error in which we are interested, namely $P_e^n = \pi_1 \beta_1(\psi_n | P_1, P_2) + \pi_2 \beta_2(\psi_n | P_1, P_2)$, where $\pi_1, \pi_2$ are the prior probabilities of each class. This shows that Gutman's test cannot assure an arbitrary level of confidence. As we will see, we can control the decay rate for the average error if we use a confidence predictor with prediction set sizes bigger than one.

**Binary Confidence Predictor - Asymptotic Results.** To build the confidence predictor, we modify the decision rule to provide a subset of the hypothesis, namely $\Gamma_n^\alpha : \mathcal{X}^{2 \times N} \times \mathcal{X}^n \to 2^{\{H_1, H_2\}}$. The error is defined as:

$$P_e^n = \mathbb{P}(H_{test} \notin \Gamma_n^\alpha(\mathbf{X}_1, \mathbf{X}_2, \mathbf{X}_{test})) \tag{13}$$

where $H_{test}$ is the hypothesis of the test sequence. We can also write:

$$P_e^n = \pi_1 \mathbb{P}(H_1 \notin \Gamma_n^\alpha(\mathbf{X}_1, \mathbf{X}_2, \mathbf{X}_{test}) | \mathbf{X}_{test}) \sim P_1) + \pi_2 \mathbb{P}(H_2 \notin \Gamma_n^\alpha(\mathbf{X}_1, \mathbf{X}_2, \mathbf{X}_{test}) | \mathbf{X}_{test}) \sim P_2),$$

where $\pi_1, \pi_2$ are prior probabilities for $H_1, H_2$. If we use Gutman's test, the error exponent for one of the conditional probabilities is controlled, namely

$$\mathbb{P}(H_1 \notin \Gamma_n^\alpha(\mathbf{X}_1, \mathbf{X}_2, \mathbf{X}_{test}) | \mathbf{X}_{test}) \sim P_1) = \beta_1(\psi_n^{\text{Gutman}} | P_1, P_2).$$

Therefore, to get the confidence guarantee, we can modify Gutman's test as follows.

**Definition 4.1.** Gutman's test with confidence is defined as follows:

- Include $H_1$ if $\text{GJS}(T_{\mathbf{X}_1}, T_{\mathbf{X}_{\text{test}}}, \alpha) < \lambda$.

- Include $H_2$ if $\text{GJS}(T_{\mathbf{X}_2}, T_{\mathbf{X}_{\text{test}}}, \alpha) < \lambda$.

The decision rule for $H_1$ is the classical Gutman's test denoted by $\psi_{1,n}^{\text{Gutman}}$, while the second rule is the same test but using $T_{\mathbf{X}_2}$ instead of $T_{\mathbf{X}_1}$, and it is denoted by $\psi_{2,n}^{\text{Gutman}}$.

We can see that the errors are related to Gutman's first error:

$$\mathbb{P}(H_1 \notin \Gamma_n^\alpha(\mathbf{X}_1, \mathbf{X}_2, \mathbf{X}_{test}) | \mathbf{X}_{test} \sim P_1) = \beta_1(\psi_{1,n}^{\text{Gutman}} | P_1, P_2) \tag{14}$$

$$\mathbb{P}(H_2 \notin \Gamma_n^\alpha(\mathbf{X}_1, \mathbf{X}_2, \mathbf{X}_{test}) | \mathbf{X}_{test} \sim P_2) = \beta_1(\psi_{2,n}^{\text{Gutman}} | P_1, P_2). \tag{15}$$

We can leverage the result from the classical Gutman's test to get a bound on the error probability.

**Theorem 4.2.** *The probability of error of Gutman's test with confidence satisfies the following:*

$$\limsup_{n \to \infty} \frac{1}{n} \log P_e^n \leq -\lambda.$$

The proof is given in Section E.1. The theorem shows the error decay rate can be controlled arbitrarily, similar to conformal prediction, but at the cost of larger or empty prediction sets. Next, we characterize the probability of larger set sizes. Let's look at the following probabilities:

$$\mathbb{P}(|\Gamma_n^\alpha(\mathbf{X}_1, \mathbf{X}_2, \mathbf{X}_{\text{test}})| = 0) = \mathbb{P}(\text{GJS}(T_{\mathbf{X}_1}, T_{\mathbf{X}_{\text{test}}}, \alpha) \geq \lambda, \text{ and GJS}(T_{\mathbf{X}_2}, T_{\mathbf{X}_{\text{test}}}, \alpha) \geq \lambda) \quad (16)$$

$$\mathbb{P}(|\Gamma_n^\alpha(\mathbf{X}_1, \mathbf{X}_2, \mathbf{X}_{\text{test}})| = 2) = \mathbb{P}(\text{GJS}(T_{\mathbf{X}_1}, T_{\mathbf{X}_{\text{test}}}, \alpha) < \lambda, \text{ and GJS}(T_{\mathbf{X}_2}, T_{\mathbf{X}_{\text{test}}}, \alpha) < \lambda) \quad (17)$$

The following theorem provides bounds on the error exponent of these probabilities.

**Theorem 4.3.** *We have:*

$$\lim_{n\to\infty} \frac{1}{n} \log \mathbb{P}(|\Gamma_n^\alpha(\mathbf{X}_1, \mathbf{X}_2, \mathbf{X}_{test})| = 0) \leq -\lambda \quad (18)$$

$$\lim_{n\to\infty} \frac{1}{n} \log \mathbb{P}(|\Gamma_n^\alpha(\mathbf{X}_1, \mathbf{X}_2, \mathbf{X}_{test})| = 2) \leq -\min\left(F(P_1, P_2, \alpha, \lambda), F(P_2, P_1, \alpha, \lambda)\right). \quad (19)$$

*where*

$$F(P_1, P_2, \alpha, \lambda) := \min_{\substack{(Q_1, Q_2) \in \mathcal{P}(\mathcal{X})^2 \\ \text{GJS}(Q_1, Q_2, \alpha) < \lambda}} D(Q_2 \| P_2) + \alpha D(Q_1 \| P_1). \quad (20)$$

*if* $\lambda > \min\left(F(P_1, P_2, \alpha, \lambda), F(P_2, P_1, \alpha, \lambda)\right)$, *the equality holds for eq. 19, otherwise for eq. 18.*

The proof is presented in Appendix E.2. The high-level idea behind the proof is as follows. Consider the case where $\mathbf{X}_{\text{test}} \sim P_1$. In this case, the probability of the event $\text{GJS}(T_{\mathbf{X}_1}, T_{\mathbf{X}_{\text{test}}}, \alpha) \geq \lambda$ decreases exponentially with probability $O(e^{-n\lambda})$. On the other hand, the probability of the event $\text{GJS}(T_{\mathbf{X}_2}, T_{\mathbf{X}_{\text{test}}}, \alpha) < \lambda$ decreases exponentially with probability $O(e^{-nF(P_1,P_2,\alpha,\lambda)})$ using Sanov's theorem. Similar arguments can be made for the case $\mathbf{X}_{\text{test}} \sim P_2$. The theorem follows from the analysis of the dominant error exponent.

The above theorem implies that it is possible to have a confidence predictor with controlled error that is asymptotically efficient, which means it yields a set of cardinality one. However, it also reveals a fundamental trade-off. As we increase $\lambda$ to have higher confidence, the decay exponent for the probability of inefficient prediction sets decreases. In the limit, if $\lambda$ is greater than $G(P_1, P_2, \alpha)$ or $G(P_2, P_1, \alpha)$, the exponent is zero, and the confidence predictor is asymptotically inefficient.

**Multi-class Confidence Predictors - Asymptotic Results.** The extension to multiple-class classification follows a similar idea. The decision function for Gutman's test with confidence is given as follows:

$$\Gamma_n^{\text{Gutman}}(\mathbf{X}_1, \ldots, \mathbf{X}_M, \mathbf{X}_{\text{test}}) = \{H_i, \forall i : \text{GJS}(T_{\mathbf{X}_i}, T_{\mathbf{X}_{\text{test}}}, \alpha) < \lambda\}. \quad (21)$$

The error probability is defined similarly as $P_e^n = \mathbb{P}\left(\mathbf{X}_{\text{test}} \notin \Gamma_n^{\text{Gutman}}(\mathbf{X}_1, \ldots, \mathbf{X}_M, \mathbf{X}_{\text{test}})\right)$. We can immediately get the following result.

**Theorem 4.4.** *The probability of error of Gutman's test with confidence for $M$ classs satisfies the following:*

$$\limsup_{n\to\infty} \frac{1}{n} \log P_e^n \leq -\lambda.$$

We do not present the proof as it is a simple extension of Theorem 4.2 given in Section E.1. Next, we characterize the probability of different set sizes. We would need to use the generalized Sanov's theorem and related analysis.

**Theorem 4.5.** *For any $k > 1$, the probability that the prediction set has the cardinality $k$ decays exponentially with the exponent bounded as follows:*

$$\lim_{n\to\infty} \frac{1}{n} \log \mathbb{P}(|\Gamma_n^{Gutman}(\mathbf{X}_1, \ldots, \mathbf{X}_M, \mathbf{X}_{test})| = k) \leq -\inf_{l\in[M]} \inf_{S\subset[M], |S|=k} F(\{P_i : i \in S\}, P_l, \alpha, \lambda)$$

*where:*

$$F(\{P_i : i \in S\}, P_l, \alpha, \lambda) := \inf_{\substack{((Q_i)_{i\in S/\{l\}}, Q_t) \in \mathcal{P}^{|S|} \\ \text{GJS}(Q_i, Q_t, \alpha) < \lambda, \forall i \in S/\{l\}}} \alpha \sum_{i\in S/\{l\}} D(Q_i \| P_i) + D(Q_t \| P_l), \quad l \in S$$

$$F(\{P_i : i \in S\}, P_l, \alpha, \lambda) := \inf_{\substack{(Q_1, \ldots, Q_M, Q_t) \in \mathcal{P}^{M+1} \\ \text{GJS}(Q_i, Q_t, \alpha) < \lambda, \forall i \in S \\ \text{GJS}(Q_l, Q_t, \alpha) \geq \lambda}} \alpha \sum_{i\in S\cup\{l\}} D(Q_i \| P_i) + D(Q_t \| P_l), \qquad l \notin S.$$

*The probability of an empty prediction set is bounded as follows, too:*

$$\lim_{n\to\infty} \frac{1}{n} \log \mathbb{P}(\left|\Gamma_n^{Gutman}(\mathbf{X}_1, \ldots, \mathbf{X}_M, \mathbf{X}_{test})\right| = 0) \leq -\lambda. \tag{22}$$

The proof is given in Section E.3. Gutman's test is known to be optimal (Gutman, 1989), in the sense that it provides the lowest type II error among all tests that uniformly control the type I error. We will discuss the implications of this optimality as well as non-asymptotic results in Appendix F.

## 5 RELATED WORKS

Conformal prediction (Vovk et al., 2022) is a framework for confidence predictors with distribution-free coverage guarantees that rely only on the assumption that the samples are exchangeable. Some notable examples are split conformal prediction (Papadopoulos et al., 2002; Lei et al., 2018), adaptive conformal prediction (Romano et al., 2020), weighted conformal prediction (Tibshirani et al., 2019; Lei & Candès, 2021) and localized conformal prediction (Guan, 2023)—see Angelopoulos & Bates (2021) for more details. Transdutive learning was introduced in Gammerman et al. (1998), while transductive conformal prediction (TCP) was proposed in Vovk (2013) to generalize conformal prediction to multiple test examples. It is in this sense that we understand transductive learning. Vovk (2013) also discussed Bonferroni predictors as an information-efficient approach to transductive prediction. For a historical anecdote on the variations on the notion of transductive learning, going back to Vapnik, see 4.8.5 in Vovk et al. (2022). Applications of transductive learning have been explored in ranking (Fermanian et al., 2025), which in itself includes many other use cases. Theoretical aspects of TCP were studied in Gazin et al. (2024), where the joint distribution of $p$-values for general exchangeable scores is derived. Although the applications of transductive learning are nascent, it provides a more general framework for studying confidence predictors.

Work on conformal prediction has focused on marginal and conditional guarantees, $p$- and $e$-value distributions, and extensions such as handling non-exchangeable samples. (Angelopoulos et al., 2024; Foygel Barber et al., 2021; Gazin et al., 2024; Vovk, 2012; Bates et al., 2023; Marques F., 2025; Vovk & Wang, 2024; 2023; Grunwald et al., 2024; Gauthier et al., 2025). In particular, two open research directions are relevant for this paper: first, the connection with hypothesis testing, and second, the theoretical bounds on the efficiency of conformal prediction.

The connection of confidence prediction with hypothesis testing is noted, for instance, in (Waudby-Smith et al., 2025; Wasserman et al., 2020). The authors in (Wasserman et al., 2020) introduced the split likelihood ratio statistics and used them to build a confidence set that enjoys finite sample guarantees and can be applied to a hypothesis testing setup. The intuitive idea is that the likelihood ratio tests, prevalent in hypothesis testing works, can be modified to build $e$-value (see eq. 6 in (Wasserman et al., 2020)). Statistical classification, on the other hand, has been seen as hypothesis testing in the past. When the distributions of each class are not given, the problem of predicting the label of new samples from a training data is seen as hypothesis testing from empirically observed statistics and was discussed in Ziv (1988) for binary classification and in Gutman (1989) for multiple hypothesis testing. The goal in these works is to characterize the optimal test and its error exponent in asymptotic (Gutman, 1989) and non-asymptotic regimes (Zhou et al., 2020; Haghifam et al., 2021). These results, however, do not address confidence prediction and assume a single output.

In (Correia et al., 2024), confidence prediction is framed as a list decoding problem in information theory. In that light, confidence predictors can be seen as list decoding for hypothesis testing with empirically observed statistics. The problem of Bayesian $M$-ary hypothesis testing with list decoding has been considered in Asadi Kangarshahi & Guillén i Fàbregas (2023), but assuming known probabilities and fixed list sizes. The problem of Bayesian $M$-ary hypothesis testing with empirically observed statistics was considered in Haghifam et al. (2021). The result is asymptotic, does not consider list decoding, and works on finite alphabets. Method of types is the primary technique for deriving bounds in the case of empirically observed statistics, which requires the assumption of a finite alphabet size. An extension to a larger alphabet has been considered in Kelly et al. (2012).

Finally, on the efficiency of conformal prediction, Correia et al. (2024) used the data processing inequality for $f$-divergences to get a lower bound on the logarithm of the expected prediction set size that mainly depends on the conditional entropy. In this work, we extend this study using a different class of bounds on hypothesis testing. Numerous information-theoretic bounds exist for

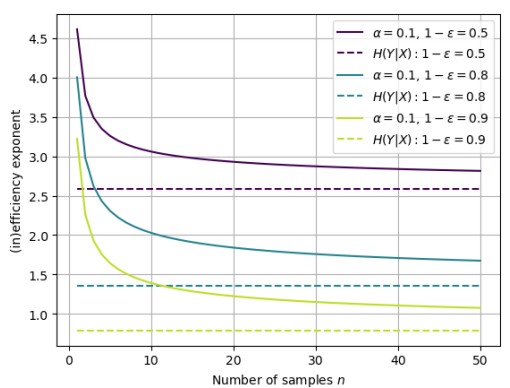
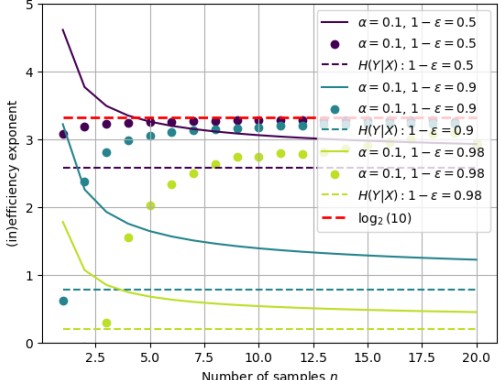

**(a)** The finite block length bound for the growth exponent of the prediction set size, namely the inefficiency versus the number of samples

**(b)** The comparison of the upper bound with naive Bonferroni split conformal prediction for transductive inference - $\alpha = 0.1$

**Figure 1:** Numerical Results for the derived theoretical bound

hypothesis testing (Verdu & Han, 1994; Han, 2014; Polyanskiy et al., 2010; Polyanskiy & Verdú, 2010; Poor & Verdu, 1995; Chen & Alajaji, 2012) with applications in finite-block-length analysis of Shannon capacity and source coding. Our bound in Theorem 3.1 generalizes Verdu & Han (1994) to variable-size list decoding; an extension to fixed-size list decoding is given in Afser (2021).

## 6   NUMERICAL RESULTS

In this section, we conduct a small experiment to illustrate the relevance of the bound. We use the MNIST dataset (LeCun et al., 1998) with noisy labels to have control over the uncertainty: each label is kept with probability $1 - \epsilon$ and changed to another class with probability $\epsilon/(N-1)$, where $N$ is the total number of classes. For this setup, we can easily compute $H(Y|X)$ and $\sigma$. We plot the efficiency rate $\gamma_{n,m}$ as a function of the number of test sample $n$ (Theorem 3.4) in Figure 1a. We plot $H(Y|X) + \sigma Q^{-1}(\alpha)/\sqrt{n} - \log_2 n/2n$, omitting constant terms $O(1)/n$ that vanish as $n$ grows. Note we use the base 2 for the logarithms in the experiments. A first observation is a persistent gap between conditional entropy (dashed line) and our bound, which closes only slowly: even for hundreds of samples, our bound provides a better guideline for the efficiency rate.

We compared our bound with a transductive method in Figure 1b. We used Bonferroni predictor as explained in Vovk (2013), which converts per-sample $p$-values to $p$-value for transductive prediction - see Section G for the details of Bonferroni predictors. For these experiments, we chose 180 samples in the calibration set to create more granularity. As it can be seen, such Bonferroni prediction becomes inefficient as $n$ increases. Particularly because the per-test confidence becomes more stringent. For example, for $\alpha = 0.1$ and $n = 20$, we need to have a confidence level of $0.005$ per sample. With limited calibration set size, this will soon get to the inefficient set prediction containing most labels. Note that our approximation can be loose for smaller $n$ because of the ignored constant terms and relaxing the assumption $\alpha + \rho/\sqrt{n}\sigma^3 + \Delta \in [0, 1]$. In long term, the impact of these terms diminishes, and our approximate lower bound holds providing a better lower bound than the conditional entropy. We provide further numerical results in App. H.

## 7   CONCLUSION

We established new theoretical bounds that rigorously characterize the trade-off between efficiency and confidence in transductive conformal prediction, offering fundamental insights into their inherent limitations. Our analysis further exposes the inefficiency of Bonferroni-based methods and underscores the need for more principled, efficient transductive predictors. Future work includes extending these bounds to exchangeable sequences and relaxing assumptions such as identical label distributions. Additionally, overcoming the reliance on the method of types and finite-alphabet settings remains a critical step toward broader applicability and practical deployment.

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

## A   ELEMENTS OF METHOD OF TYPES AND LARGE DEVIATION

Consider a finite space $\mathcal{X}$, and a random variables $X_i \sim P$ where $P \in \mathcal{P}(\mathcal{X})$. The type of a sequence $\mathbf{X} = (X_1, \ldots, X_n)$ is defined as

$$T_{\mathbf{X}}(a) := \frac{1}{n} \sum_{i=1}^{n} \mathbf{1}(X_i = a), a \in \mathcal{X}.$$

This is the empirical distribution of the sequence. The set of type $P$ is defined as:

$$T(P) := \{\mathbf{X} \in \mathcal{X}^n : T_{\mathbf{x}} = P\}.$$

Also the set of all types for sequences of length $n$ is denoted by $\mathcal{P}_n$. We summarize the main properties of types in the following theorem. The proof can be found in (Csiszár & Körner, 2011; Dembo & Zeitouni, 2009).

**Theorem A.1.** *Consider a finite space $\mathcal{X}$ with the set of all probability distributions over $\mathcal{X}$ denoted by $\mathcal{P}$, and the set of all types of the sequences of length $n$ denoted $\mathcal{P}_n$. By $P(A), Q(A), \ldots$, we denote the probability of set $A$ according to the probability measure $P, Q, \ldots$, and we assume that for $A \subset \mathcal{X}^n$ we implicitly use the product measure. We have the following properties for types.*

- *$|\mathcal{P}_n| \leq (n+1)^{|\mathcal{X}|}$*

- *for all $\mathbf{X} \in T(P)$: $P(\mathbf{X}) = e^{-nH(P)}$*

- *for all $\mathbf{X} \in \mathcal{X}^n$ and $P \in \mathcal{P}$: $P(\mathbf{X}) = e^{-n(H(T_{\mathbf{x}}) + D(T_{\mathbf{x}}\|P))}$*

- *for all $P \in \mathcal{P}_n$: $\frac{1}{(n+1)^{|\mathcal{X}|}} e^{nH(P)} \leq |T(P)| \leq e^{nH(P)}$*

- *for all $P \in \mathcal{P}_n$ and $Q \in \mathcal{P}$: $\frac{1}{(n+1)^{|\mathcal{X}|}} e^{-nD(P\|Q)} \leq Q\left(T(P)\right) \leq e^{-nD(P\|Q)}$,*

*where $H(P) = -\sum_{x \in \mathcal{X}} P(x) \log P(x)$ is the Shannon entropy of $P$, and $D(P\|Q)$ is the Kullback-Leibler divergence.*

In many cases, we are interested in establishing a bound on the decay exponent of certain probabilities involving types. The following result from large deviation theory is the key tool for such derivations. See (Dembo & Zeitouni, 2009) for more details.

**Theorem A.2** (Sanov's theorem). *For any set $\Gamma^\alpha \in \mathcal{P}(\mathcal{X})$, and any random sequence $\mathbf{X} \in \mathcal{X}^n$ drawn i.i.d. using $P$, we have:*

$$-\inf_{Q \in \text{int}(\Gamma^\alpha)} D(Q\|P) \leq \liminf_{n \to \infty} \frac{1}{n} \log P(T_{\mathbf{X}} \in \Gamma^\alpha) \leq \limsup_{n \to \infty} \frac{1}{n} \log P(T_{\mathbf{X}} \in \Gamma^\alpha) \leq -\inf_{Q \in \Gamma^\alpha} D(Q\|P),$$
(23)

*where $\text{int}(\Gamma^\alpha)$ is the interior of the set $\Gamma^\alpha$. In particular for any set $\Gamma^\alpha$ whose closure contains its interior, we have:*

$$\lim_{n \to \infty} \frac{1}{n} \log P(T_{\mathbf{X}} \in \Gamma^\alpha) = -\inf_{Q \in \Gamma^\alpha} D(Q\|P).$$

We provide a general version of this theorem with its proof to be self-contained. This version will be directly useful for our results.

**Theorem A.3.** *Consider the sequences $\mathbf{X}_i \in \mathcal{X}^{N_i}$ drawn i.i.d. from $P_i$ for $i \in [M]$ with $N_i = \alpha_i n$ for $\alpha_i \in [0, 1]$, and the types of these sequences are denoted by $T_{\mathbf{X}_i}$. Then for any set of probability distributions $\Omega \subset \mathcal{P}^M$, we have:*

$$-\inf_{(Q_1, \ldots, Q_M) \in \text{int}(\Omega)} \sum_{i=1}^{M} \alpha_i D(Q_i\|P_i) \leq \liminf_{n \to \infty} \frac{1}{n} \log \mathbb{P}((T_{\mathbf{X}_1}, \ldots, T_{\mathbf{X}_M}) \in \Omega)$$

$$\leq \limsup_{n \to \infty} \frac{1}{n} \log \mathbb{P}((T_{\mathbf{X}_1}, \ldots, T_{\mathbf{X}_M}) \in \Omega) \leq -\inf_{(Q_1, \ldots, Q_M) \in \Omega} \sum_{i=1}^{M} \alpha_i D(Q_i\|P_i)$$
(24)

*where $\text{int}(\Omega)$ is the interior of $\Omega$. Besides, if the closure of $\Omega$ contains its interior, we have the equality.*

*Proof.* We first establish an upper bound using properties of types as follows:

$$\mathbb{P}((T_{\mathbf{X}_1}, \ldots, T_{\mathbf{X}_M}) \in \Omega) = \sum_{(T_{\mathbf{X}_1}, \ldots, T_{\mathbf{X}_M}) \in \Omega \cap \mathcal{P}_n^M} \prod_{i=1}^{M} P_i(T_{\mathbf{X}_i})$$

$$\leq \sum_{(T_{\mathbf{X}_1}, \ldots, T_{\mathbf{X}_M}) \in \Omega \cap \mathcal{P}_n^M} \prod_{i=1}^{M} e^{-N_i D(T_{\mathbf{X}_i} \| P_i)}$$

$$\leq \prod_{i=1}^{M} (N_i + 1)^{|\mathcal{X}|} \exp\left(-n \inf_{(T_{\mathbf{X}_1}, \ldots, T_{\mathbf{X}_M}) \in \Omega \cap \mathcal{P}_n^M} \sum_{i=1}^{M} \alpha_i D(T_{\mathbf{X}_i} \| P_i)\right)$$

Next, we focus on the lower bound using similar techniques:

$$\mathbb{P}((T_{\mathbf{X}_1}, \ldots, T_{\mathbf{X}_M}) \in \Omega) = \sum_{(T_{\mathbf{X}_1}, \ldots, T_{\mathbf{X}_M}) \in \Omega \cap \mathcal{P}_n^M} \prod_{i=1}^{M} P_i(T_{\mathbf{X}_i})$$

$$\geq \sum_{(T_{\mathbf{X}_1}, \ldots, T_{\mathbf{X}_M}) \in \Omega \cap \mathcal{P}_n^M} \prod_{i=1}^{M} \frac{1}{(N_i + 1)^{|\mathcal{X}|}} e^{-N_i D(T_{\mathbf{X}_i} \| P_i)}$$

$$\geq \prod_{i=1}^{M} \frac{1}{(N_i + 1)^{|\mathcal{X}|}} e^{-N_i D(T_{\mathbf{X}_i} \| P_i)} \text{ for all } (T_1, T_2, \ldots, T_M) \in \Omega \cap \mathcal{P}_n^M$$

$$\geq \prod_{i=1}^{M} \frac{1}{(N_i + 1)^{|\mathcal{X}|}} \exp\left(-n \inf_{(T_{\mathbf{X}_1}, \ldots, T_{\mathbf{X}_M}) \in \Omega \cap \mathcal{P}_n^M} \sum_{i=1}^{M} \alpha_i D(T_{\mathbf{X}_i} \| P_i)\right)$$

Using the fact that $\lim_{n \to \infty} \frac{1}{n}(n + 1)^{|\mathcal{X}|} = 0$, we get the following equalities:

$$\limsup_{n \to \infty} \frac{1}{n} \log \mathbb{P}((T_{\mathbf{X}_1}, \ldots, T_{\mathbf{X}_M}) \in \Omega) = -\liminf_{n \to \infty} \inf_{(T_{\mathbf{X}_1}, \ldots, T_{\mathbf{X}_M}) \in \Omega \cap \mathcal{P}_n^M} \sum_{i=1}^{M} \alpha_i D(T_{\mathbf{X}_i} \| P_i) \quad (25)$$

$$\liminf_{n \to \infty} \frac{1}{n} \log \mathbb{P}((T_{\mathbf{X}_1}, \ldots, T_{\mathbf{X}_M}) \in \Omega) = -\limsup_{n \to \infty} \inf_{(T_{\mathbf{X}_1}, \ldots, T_{\mathbf{X}_M}) \in \Omega \cap \mathcal{P}_n^M} \sum_{i=1}^{M} \alpha_i D(T_{\mathbf{X}_i} \| P_i) \quad (26)$$

Then since $\Omega \cap \mathcal{P}_n^M \subseteq \Omega$, the upper bound is obvious. For the lower bound, we start from the fact for any point $\mu$ in the interior of $\Omega$ having the same support as $P_1 \times \cdots \times P_M$, we can find a distribution in $\mathcal{P}_n$ with the total variation distance of at most $O(1/n)$ from $\mu$ (see Lemma 2.1.2 of (Dembo & Zeitouni, 2009)). Therefore, we have a sequence of distributions in $\Omega \cap \mathcal{P}_n$ that converges to $\mu$. Using this sequence, we can get a lower bound for each $\mu$ in the interior of $\Omega$, which proves the final lower bound. $\square$

## B    PROOFS OF SECTION 3

The basic idea for deriving the new inequalities is based on a set of results in information theory and Shannon's channel coding theorem. In Shannon Theory, it is known that Fano's inequality cannot be used to prove strong converse results and establish the phase transition at the Shannon capacity. Other inequalities in information theory involve the underlying probability distribution or information density terms (Verdu & Han, 1994; Han, 2014). For example, Theorem 4 in (Verdu & Han, 1994) states that:

$$\mathbb{P}(P(X|Y) \leq \beta) \leq \epsilon + \beta,$$

where $\epsilon$ is the error probability of a code for a channel $P_{Y|X}$ and $\beta$ is an arbitrary number in $[0, 1]$. These bounds can sometimes lead to tighter results compared to Fano's inequality, as indicated in (Polyanskiy et al., 2010). In what follows, we obtain a similar bound for transductive conformal prediction.

## B.1 PROOF OF THEOREM 3.1

*Proof.* To simplify the notation for the proof, we denote $\boldsymbol{X} := X_{m+1}^{m+n}$, $\boldsymbol{Y} := Y_{m+1}^{m+n}$ and $\boldsymbol{Z} := Z_1^m$. we assume $(\boldsymbol{X}, \boldsymbol{Y})$ are drawn from the distribution $P$. For each $\boldsymbol{X} \in \mathcal{X}^n$, define the set:

$$B_{\boldsymbol{X}} = \{\boldsymbol{Y} : P(\boldsymbol{Y}|\boldsymbol{X}) \le \beta\}.$$

$P(\boldsymbol{Y}|\boldsymbol{X})$ is the conditional distribution induced by $P$. The proof follows the steps below:

$$\mathbb{P}(P(\boldsymbol{Y}|\boldsymbol{X}) \le \beta) = \int P(B_{\boldsymbol{X}}|\boldsymbol{X})P(\boldsymbol{X})d\boldsymbol{X} = \int P(\boldsymbol{X}, B_{\boldsymbol{X}})d\boldsymbol{X} = \int P(\boldsymbol{X}, \boldsymbol{Z}, B_{\boldsymbol{X}})d\boldsymbol{X}d\boldsymbol{Z}$$

$$= \int \int P(\boldsymbol{X}, \boldsymbol{Z}, B_{\boldsymbol{X}} \cap \Gamma^\alpha(\boldsymbol{Z}, \boldsymbol{X})^c)d\boldsymbol{X}d\boldsymbol{Z} + \int \int P(\boldsymbol{X}, \boldsymbol{Z}, B_{\boldsymbol{X}} \cap \Gamma^\alpha(\boldsymbol{Z}, \boldsymbol{X}))d\boldsymbol{X}d\boldsymbol{Z}$$

$$\le \int \int P(\boldsymbol{X}, \boldsymbol{Z}, \Gamma^\alpha(\boldsymbol{Z}, \boldsymbol{X})^c)d\boldsymbol{X}d\boldsymbol{Z} + \int \int P(\boldsymbol{X}, \boldsymbol{Z}, B_{\boldsymbol{X}} \cap \Gamma^\alpha(\boldsymbol{Z}, \boldsymbol{X}))d\boldsymbol{X}d\boldsymbol{Z}$$

$$\stackrel{(1)}{\le} \alpha + \int \int P(\boldsymbol{X}, \boldsymbol{Z}, B_{\boldsymbol{X}} \cap \Gamma^\alpha(\boldsymbol{Z}, \boldsymbol{X}))d\boldsymbol{X}d\boldsymbol{Z}$$

$$= \alpha + \int \int P(\boldsymbol{X}, \boldsymbol{Z})P(B_{\boldsymbol{X}} \cap \Gamma^\alpha(\boldsymbol{Z}, \boldsymbol{X})|\boldsymbol{X}, \boldsymbol{Z})d\boldsymbol{X}d\boldsymbol{Z}$$

$$= \alpha + \int \int P(\boldsymbol{X}, \boldsymbol{Z}) \left( \sum_{\boldsymbol{Y}, \boldsymbol{Y} \in B_{\boldsymbol{X}} \cap \Gamma^\alpha(\boldsymbol{Z}, \boldsymbol{X})} P(\boldsymbol{Y}|\boldsymbol{X}, \boldsymbol{Z}) \right) d\boldsymbol{X}d\boldsymbol{Z}$$

$$\stackrel{(2)}{\le} \alpha + \int \int P(\boldsymbol{X}, \boldsymbol{Z}) \left( \sum_{\boldsymbol{Y}, \boldsymbol{Y} \in B_{\boldsymbol{X}} \cap \Gamma^\alpha(\boldsymbol{Z}, \boldsymbol{X})} \beta \right) d\boldsymbol{X}d\boldsymbol{Z}$$

$$\le \alpha + \int P(\boldsymbol{X}, \boldsymbol{Z})\beta|\Gamma^\alpha(\boldsymbol{Z}, \boldsymbol{X})|d\boldsymbol{X}d\boldsymbol{Z} = \alpha + \beta\mathbb{E}(|\Gamma^\alpha(\boldsymbol{Z}, \boldsymbol{X})|).$$

The inequality (1) follows from the confidence assumption:

$$P\left(Y_{m+1}^{m+n} \notin \Gamma^\alpha(Z_1^m, X_{m+1}^{m+n})\right) = \int \int P(\boldsymbol{X}, \boldsymbol{Z}, \Gamma^\alpha(\boldsymbol{Z}, \boldsymbol{X})^c)d\boldsymbol{X}d\boldsymbol{Z} \le \alpha.$$

The inequality (2) follows from the independence of $\boldsymbol{Z}$ and $(\boldsymbol{X}, \boldsymbol{Y})$ and the definition of $B_{\boldsymbol{X}}$. $\qquad \square$

*Remark* B.1. Note that Theorem 3.1 can be written as:

$$\sup_{\beta \in [0,1]} \frac{\mathbb{P}(P(Y_{m+1}^{m+n}|X_{m+1}^{m+n}) \le \beta) - \alpha}{\beta} \le \mathbb{E}(|\Gamma^\alpha(Z_1^m, X_{m+1}^{m+n})|) \tag{27}$$

This bound means that the prediction set size is large even if for a single $\beta$, the left hand side is large. As an example, suppose that the conditional probability distribution is not concentrated around a point and has large spread. In other words, the uncertainty is high, and we can find many labels with equally high but numerically small probabilities. In this case, we can choose a small $\beta$ that yields a very high probability of $P(Y_{m+1}^{m+n}|X_{m+1}^{m+n}) \le \beta$. Therefore, the prediction set size scales mainly with with $1/\beta$ and is expected to be large as $\beta$ is small. Intuitively, the left hand side of Theorem 3.1 measures the intrinsic uncertainty and right hand side measures the prediction set size.

## B.2 PROOF OF THEOREM 3.2

*Proof.* We use Theorem 3.1 to prove the results. The choice of $\beta$ can be important. Let's choose $\beta = e^{-n(H(Y|X)-\delta)}$ where $H(Y|X)$ is the conditional entropy, and $\delta$ is any non-negative number. With standard manipulations, we obtain the following result from Theorem 3.1:

$$\mathbb{P}\left(\frac{1}{n}\log P(Y_{m+1}^{m+n}|X_{m+1}^{m+n}) \le -H(Y|X) + \delta\right) \le \alpha_n + e^{-n(H(Y|X)-\delta)}\mathbb{E}(|\Gamma^\alpha(Z_1^m, X_{m+1}^{m+n})|).$$

$$\tag{28}$$

Since the test samples are i.i.d., the term $\log P(Y_{m+1}^{m+n}|X_{m+1}^{m+n})$ can be decomposed as:

$$\frac{1}{n} \log P(Y_{m+1}^{m+n}|X_{m+1}^{m+n}) = \frac{1}{n} \sum_{i=1}^{n} \log P(Y_{m+i}|X_{m+i}).$$

From law of large numbers, as $n$ goes to infinity, the sum converges almost surely to the negative conditional entropy between the input $X$ and the label $Y$, namely $-H(Y|X)$. This means that the probability on the left hand side of eq. 28 goes to one. We have two cases:

- Case 1: if $\gamma_m^- < H(Y|X)$, then we have:

$$\liminf_{n\to\infty} e^{-n(H(Y|X)-\delta)} \mathbb{E}(|\Gamma^\alpha(Z_1^m, X_{m+1}^{m+n})|) = 0.$$

  This means that $\lim_{n\to\infty} \alpha_n = 1$, which means the confidence goes to zero.

- Case 2: for non-trivial asymptotic confidence, $\liminf_{n\to\infty} \alpha_n$ is strictly below one. For the inequality to hold, the second term needs to be non-vanishing, that is $\gamma_m^- > H(Y|X) - \delta$, namely $\gamma_m^- \geq H(Y|X)$.

The proof follows accordingly. $\square$

### B.3  PROOF OF THEOREM 3.4

*Proof.* We use Berry-Esseen central limit theorem for the proof.

**Theorem B.2** (Berry-Esseen). *Let $X_i$, $i \in [n]$ be i.i.d. random variables with $\mathbb{E}(X_i) = \mu$, $Var(X_i) = \sigma^2$, $\rho = \mathbb{E}(|X_i - \mu|^3)$. Then we have for any $t \in \mathbb{R}$:*

$$\left| \mathbb{P}\left( \frac{1}{\sqrt{n}} \sum_{i=1}^{n} \frac{(X_i - \mu)}{\sigma} \geq t \right) - Q(t) \right| \leq \frac{\rho}{\sqrt{n}\sigma^3}.$$

We use Theorem 3.1 as starting point:

$$\mathbb{P}\left( P(Y_{m+1}^{m+n}|X_{m+1}^{m+n}) < \beta \right) = \mathbb{P}\left( \frac{1}{\sqrt{n}} \log P(Y_{m+1}^{m+n}|X_{m+1}^{m+n}) \leq \frac{1}{\sqrt{n}} \log \beta \right)$$

$$= \mathbb{P}\left( \frac{1}{\sqrt{n}} \sum_{i=1}^{n} \log P(Y_{m+i}|X_{m+i}) \leq \frac{1}{\sqrt{n}} \log \beta \right)$$

$$= \mathbb{P}\left( \frac{1}{\sqrt{n}} \sum_{i=1}^{n} \frac{(\log P(Y_{m+i}|X_{m+i}) - \mu)}{\sigma} \leq \frac{1}{\sigma\sqrt{n}} \log \beta - \frac{\sqrt{n}\mu}{\sigma} \right)$$

$$\geq Q\left( \frac{-1}{\sigma\sqrt{n}} \log \beta + \frac{\sqrt{n}\mu}{\sigma} \right) - \frac{\rho}{\sqrt{n}\sigma^3}$$

where $\mu = -H(Y|X)$ and the last step follows from Berry-Esseen. Now choose $\beta = \exp\left( n\mu - Q^{-1}(\epsilon)\sigma\sqrt{n} \right)$, which implies that:

$$Q\left( \frac{-1}{\sigma\sqrt{n}} \log \beta + \frac{\sqrt{n}\mu}{\sigma} \right) = \epsilon$$

We get the following simplified inequality for any $\epsilon$:

$$\epsilon \leq \alpha + \frac{\rho}{\sqrt{n}\sigma^3} + \exp\left( n\mu - Q^{-1}(\epsilon)\sigma\sqrt{n} + n\gamma_{n,m} \right)$$

choose $\epsilon = \alpha + \frac{\rho}{\sqrt{n}\sigma^3} + \Delta$ for any $\Delta > 0$, such that $\epsilon \in (0, 1)$. Then, we get the result:

$$\log \Delta - n\mu + Q^{-1}(\alpha + \frac{\rho}{\sqrt{n}\sigma^3} + \Delta)\sigma\sqrt{n} \leq n\gamma_{n,m}.$$

$\square$

**Deriving An Approximate Bound.** Note that $Q(x)$ is non-increasing and $(1/\sqrt{2\pi})-$Lipschitz (given that $Q'(x)$ is negative Gaussian density function). Therefore, we have:

$$\mathrm{Q}^{-1}\left(\alpha + \frac{\rho}{\sqrt{n}\sigma^3} + \Delta\right) \geq \mathrm{Q}^{-1}\left(\alpha\right) - \frac{1}{\sqrt{2\pi}}\left(\frac{\rho}{\sqrt{n}\sigma^3} + \Delta\right),$$

We can choose $\Delta = \Delta'/\sqrt{n}$, and show that there is a constant $C_0$ such that:

$$\mathrm{Q}^{-1}\left(\alpha + \frac{\rho}{\sqrt{n}\sigma^3} + \Delta\right) \geq \mathrm{Q}^{-1}\left(\alpha\right) - \frac{C_0}{\sqrt{n}}.$$

which gives the approximate exponent:

$$\log \Delta' - C_0\sigma - \frac{1}{2}\log n - n\mu + Q^{-1}(\alpha)\sigma\sqrt{n} \leq n\gamma_{n,m}.$$

The term $\log \Delta' - C_0\sigma$ is constant and hence $O(1)$. The final result follows as:

$$n\gamma_{n,m} \geq nH(Y|X) + \sqrt{n}\sigma\mathrm{Q}^{-1}\left(\alpha\right) - \frac{\log n}{2} + O(1).$$

*Remark* B.3. Looking at the proof more closely, a precise statement of the bound is as follows:

$$\epsilon \leq \alpha + \frac{\rho}{\sqrt{n}\sigma^3} + \exp\left(n\mu - Q^{-1}(\epsilon)\sigma\sqrt{n} + n\gamma_{n,m}\right).$$

The choice of $\epsilon = \alpha + \frac{\rho}{\sqrt{n}\sigma^3} + \Delta$ for any $\Delta > 0$ needs to satisfy $\epsilon \in (0,1)$. Our approximation ignores this condition, which can lead to vacuous results. In certain cases, even $\alpha + \frac{\rho}{\sqrt{n}\sigma^3}$ can be outside $(0,1)$, which yields a vacuous bound, although asymptotically as $n \to \infty$, the term will always be within the desired range. Another component is $\Delta$, which is between $(0,1)$. This means that $\log \Delta < 0$, and therefore, the actual bound is smaller that $nH(Y|X) + \sqrt{n}\sigma\mathrm{Q}^{-1}\left(\alpha\right)$. Again, as $n$ increases, these impacts vanish, and the bound should be non-vacuous.

# C  EFFICIENCY-CONFIDENCE TRADE-OFF FOR GENERAL NOTIONS OF EFFICIENCY

In (Vovk et al., 2022, Section 3.1), various criteria for efficiency has been discussed such as sum, number, unconfidence, fuzziness, multiple, and excess criterion. Our notion of efficiency based on the prediction set size is the number criterion in the transductive setting. Here. we can generalize the result for a general criterion of efficiency that can be expressed by a measure (not necessarily a probability measure). We start with the following more general result.

**Theorem C.1.** *Consider a transductive conformal predictor* $\Gamma^\alpha(Z_1^m, X_{m+1}^{m+n})$ *given a labeled dataset* $Z_1^m$ *and test samples* $X_{m+1}^{m+n}$ *with unknown labels* $Y_{m+1}^{m+n}$*. If the predictor has the confidence* $1 - \alpha$*, then for any positive* $\beta$ *and any measure Q, we have:*

$$\mathbb{P}(P(Y_{m+1}^{m+n}|X_{m+1}^{m+n}) \leq \beta Q(Y_{m+1}^{m+n}|X_{m+1}^{m+n})) \leq$$

$$\alpha + \beta \int P(X_{m+1}^{m+n}, Z_1^m)Q(\Gamma^\alpha(Z_1^m, X_{m+1}^{m+n})|X_{m+1}^{m+n})dX_{m+1}^{m+n}dZ_1^m.$$

*Proof.* We follow the idea of the proof given in B.1. Define:

$$B_{\boldsymbol{X}} := \{\boldsymbol{Y} : P(\boldsymbol{Y}|\boldsymbol{X}) \leq \beta Q(\boldsymbol{Y}|\boldsymbol{X})\}.$$

We need to modify the last step the of the proof as follows:

$$\mathbb{P}(P(\boldsymbol{Y}|\boldsymbol{X}) < \beta Q(\boldsymbol{Y}|\boldsymbol{X})) = \int P(\boldsymbol{X}, B_{\boldsymbol{X}})d\boldsymbol{X}$$

$$\leq \alpha + \int\int P(\boldsymbol{X}, \boldsymbol{Z})\left(\sum_{\boldsymbol{Y}, \boldsymbol{Y} \in B_{\boldsymbol{X}} \cap \Gamma^\alpha(\boldsymbol{Z}, \boldsymbol{X})} \beta Q(\boldsymbol{Y}|\boldsymbol{X})\right)d\boldsymbol{X}d\boldsymbol{Z}$$

$$\leq \alpha + \beta \int P(\boldsymbol{X}, \boldsymbol{Z})Q(\Gamma^\alpha(\boldsymbol{Z}, \boldsymbol{X})|\boldsymbol{X})d\boldsymbol{X}d\boldsymbol{Z}.$$

$\square$

Finally, we can extend the result to the non-asymptotic case. The measure $Q$ can be the one represented by the model or can be any notion of efficiency as before. A similar trick has been used in Eq. (102) of (Polyanskiy et al., 2010) in their meta-converse analysis.

**Theorem C.2.** *For a transduction conformal predictor with confidence $1 - \alpha$. Define:*

$$\gamma_{n,m}^{(Q)} := \frac{1}{n} \log \int P(X_{m+1}^{m+n}, Z_1^m) Q(\Gamma^\alpha(Z_1^m, X_{m+1}^{m+n}) | X_{m+1}^{m+n}) dX_{m+1}^{m+n} dZ_1^m$$

$$= \frac{1}{n} \log \mathbb{E}_{X_{m+1}^{m+n}, Z_1^m} \left( Q(\Gamma^\alpha(Z_1^m, X_{m+1}^{m+n}) | X_{m+1}^{m+n}) \right),$$

*for any measure $Q(Y|X)$ satisfying $Q(Y_{m+1}^{m+n} | X_{m+1}^{m+n}) = \prod_{i=1}^n Q(Y_{m+i} | X_{m+i})$. Then for any $n$ and $\Delta > 0$ such that $\alpha + \frac{\rho}{\sqrt{n}\sigma^3} + \Delta \in [0, 1]$, we have:*

$$\log \Delta + n\mu + \sqrt{n}\sigma Q^{-1} \left( \alpha + \frac{\rho}{\sqrt{n}\sigma^3} + \Delta \right) \le n\gamma_{n,m}^{(Q)}$$

*where $Q(\cdot)$ is the Q-function, and:*

$$\mu := \mathbb{E} \left( \log \frac{P(Y|X)}{Q(Y|X)} \right) \tag{29}$$

$$\sigma := \mathrm{Var} \left( \log \frac{P(Y|X)}{Q(Y|X)} \right)^{1/2} = \left( \mathbb{E} \left( \log \frac{P(Y|X)}{Q(Y|X)} - \mu \right)^2 \right)^{1/2} \tag{30}$$

$$\rho := \mathbb{E} \left( \left| \log \frac{P(Y|X)}{Q(Y|X)} - \mu \right|^3 \right). \tag{31}$$

The proof follows the exact same steps as in the proof given B.3, and we omit it.

Note that if $Q(\cdot)$ is a probability measure, the term $\mu$ is given by $\mathbb{E}\left( D_{KL}\left( P(Y|X) \| Q(Y|X) \right) | X \right)$. One insight from the above theorem is that the exponent of the transductive prediction efficiency, measured using a probability measure, is asymptotically the KL-divergence between the used measure and ground truth conditional probability.

**Other efficiency metrics.** Various other efficiency metrics are used in the literature. Most of these are discussed in Chapter 3 of Vovk et al. (2022). Two categories are noteworthy. One is called observed criteria where the efficiency is measured based on the observation of the data label. The other category of criteria measures the efficiency using the prediction set only. Our derivation here can be applied to observed criterion of efficiency assuming the ratios are well defined. We consider some of these measures here.

Consider first N-criterion (N for number), defined as:

$$Q(\Gamma^\alpha(X_{m+1}^{m+n})) = \frac{1}{n} \sum_{i=1}^n \left| \Gamma_i^\alpha(X_{m+1}^{m+n}) \right|,$$

where $\Gamma_i^\alpha(X_{m+1}^{m+n})$ is defined as the set of different labels predicted in $\Gamma_i^\alpha(X_{m+1}^{m+n})$ for the sample $m + i$. For this choice of $Q$, we get:

$$Q(Y_{m+1}^{m+n} | X_{m+1}^{m+n}) = \frac{1}{n} \sum_{i=1}^n 1 = 1.$$

This means that for this choice, it is still the best to simply threshold the conditional probability. The fundamental bounds on the efficiency rate, therefore, remains very similar.

Next, consider S-criterion (S for sum). This is defined as sum of $p$-values for all the labels across test samples. We consider a modified version defined as:

$$Q(\Gamma^\alpha(X_{m+1}^{m+n})) = \sum_{y_{m+1}^{m+n} \in \Gamma^\alpha(X_{m+1}^{m+n})} p_{y_{m+1}^{m+n}},$$

where $p_{y_{m+1}^{m+n}}$ is computed given the calibration set $Z_1^m$ and $X_{m+1}^{m+n}$. In this case, we have:

$$Q(Y_{m+1}^{m+n}|X_{m+1}^{m+n}) = p(Y_{m+1}^{m+n}|X_{m+1}^{m+n}),$$

where we made the conditioning of $p$-value on the test data explicit in the notation. In this case, the fundamental limits will be determined by the following ration:

$$\log \frac{P(Y|X)}{p(Y|X)},$$

where the small $p$ represents the $p$-value of $Y$ given $X$. $p$-values are between 0 and 1 but not a probability measure.

We introduce a third efficiency criterion called R-criterion (R for risk). This notion measures the risk of the labels in the prediction set. Let's consider the autonomous driving use case and the object detection application. Different objects can lead to different course of actions, each incurring different costs. Therefore, we might want to measure the average risk incurred by the prediction set using a risk measure. This application corresponds to the weighted set size as the efficiency measure where the weight of each label is proportional to its risk. We define this risk as:

$$R(\Gamma^\alpha(X_{m+1}^{m+n})) = \sum_{y \in \Gamma^\alpha(X_{m+1}^{m+n})} R(y).$$

We leave the risk function $R(\cdot)$ quite general so it can apply to a single label or a sequence of labels. We can replace $Q$ with $R$ in the above result. Note that $R(Y|X) = R(Y)$. The fundamental limits, in this case, are the moments of the following ratio:

$$\log \frac{P(Y|X)}{R(Y)}.$$

In other words, the conditional probability needs to be scaled with the risk function for optimal performance.

## D    DISCUSSION ON ACHIEVABILITY ON NON-ASYMPTOTIC BOUNDS ON EFFICIENCY

*Proof.* We start with the following lemma, which gives a bound on the expected set size.

**Lemma D.1.** *Consider two spaces for $\mathcal{X}$ and $\mathcal{Y}$ with a joint probability distribution $P(x, y)$ defined over the product space for $x \in \mathcal{X}$ and $y \in \mathcal{Y}$. Define the set $A_x$ for $x \in \mathcal{X}$ as follows:*

$$A_x = \{y : P(y|x) \geq \beta\}.$$

*Then:*

$$\mathbb{E}_X[|A_X|] \leq \frac{1}{\beta}.$$

*Proof.* The proof is as follows:

$$\mathbb{P}(A_X) = \sum_{(x,y) \in \mathcal{X} \times \mathcal{Y}} P(x, y)\mathbf{1}(y \in A_x) \tag{32}$$

$$\geq \sum_{(x,y) \in \mathcal{X} \times \mathcal{Y}} P(x)\beta\mathbf{1}(Y \in A_x) \tag{33}$$

$$= \sum_{x \in \mathcal{X}} P(x) \sum_{y \in A_x} \beta\mathbf{1}(Y \in A_x) \tag{34}$$

$$= \beta\mathbb{E}[|A_X|], \tag{35}$$

where we used to inequality $\mathbb{P}(y|x) \geq \beta$ for $y \in A_x$. Using $\mathbb{P}(A_X) \leq 1$, we get the inequality. □

Now, we just need to pick $\beta$ such that the probability of $A_X$ satisfies the required confidence level. To do so, consider the set of labels:

$$\Gamma^\alpha(x_1^n) := \{y_1^n : P(y_1^n|x_1^n) \geq \beta\}.$$

When $(X_i, Y_i)$ are independently and identically drawn from $P(X, Y)$, we can use the Berry-Esseen central limit theorem, Theorem B.2, to bound the probability of the set $\Gamma^\alpha(x_1^n)$. The probability of error is the probability that the labels $Y_{m+1}^{m+n}$ do not belong to the set $\Gamma^\alpha(X_1^n)$. It can be bounded as follows.

$$\mathbb{P}\left(P(Y_1^n|X_1^n) \leq \beta\right) = \mathbb{P}\left(\frac{1}{\sqrt{n}} \sum_{i=1}^{n} \frac{(\log P(Y_i|X_i) - \mu)}{\sigma} \leq \frac{1}{\sigma\sqrt{n}} \log \beta - \frac{\sqrt{n}\mu}{\sigma}\right)$$

$$\leq Q\left(\frac{-1}{\sigma\sqrt{n}} \log \beta + \frac{\sqrt{n}\mu}{\sigma}\right) + \frac{\rho}{\sqrt{n}\sigma^3}.$$

As before $\mu = -H(Y|X)$, $\sigma$ is the variance of the log probability, and the last step follows from Berry-Esseen theorem. To guarantee the confidence level $\alpha$, we need to choose $\beta$ as follows:

$$Q\left(\frac{-1}{\sigma\sqrt{n}} \log \beta + \frac{\sqrt{n}\mu}{\sigma}\right) + \frac{\rho}{\sqrt{n}\sigma^3} = \alpha$$

which yields the following choice of $\beta$

$$\beta = \exp\left(n\mu - Q^{-1}(\alpha - \frac{\rho}{\sqrt{n}\sigma^3})\sigma\sqrt{n}\right)$$

This is conditioned on $\alpha - \frac{\rho}{\sqrt{n}\sigma^3} \in (0, 1)$, which might not hold for smaller $n$. Indeed, we need to have:

$$n > \left(\frac{\rho}{\alpha\sigma^3}\right)^2.$$

The function $Q^{-1}(\cdot)$ is $1/\sqrt{2\pi}$-Lipschitz, and we have:

$$Q^{-1}(\alpha - \frac{\rho}{\sqrt{n}\sigma^3}) \leq Q^{-1}(\alpha) + \frac{1}{2\pi} \frac{\rho}{\sqrt{n}\sigma^3}.$$

Therefore:

$$\beta \leq \exp\left(n\mu - \sigma\sqrt{n}Q^{-1}(\alpha) + \frac{1}{2\pi} \frac{\rho}{\sigma^2}\right)$$

With this choice of $\beta$, the expected set size is bounded using the above lemma as:

$$\mathbb{E}[|\Gamma^\alpha(X_1^n)|] \leq \exp\left(-n\mu + \sigma\sqrt{n}Q^{-1}(\alpha) - \frac{1}{2\pi} \frac{\rho}{\sigma^2},\right)$$

which yields the result. $\qquad\square$

Our derivation does not contain the logarithmic terms, $-\frac{1}{2}\log n$ that appears in the lower bound. We can use a different technique, similar to the one used in (Kontoyiannis & Verdu, 2014) for the lossless compression case, to get this term as well.

**Achievability for approximate distributions.** Suppose that we have access to the approximation of the conditional distribution $P(y|X)$, given by $Q(y|x)$, and the transductive confidence sets are constructed using $Q$ as follows:

$$\Gamma_Q^\alpha(x_1^n) := \{y_1^n : Q(y_1^n|x_1^n) \geq \beta\}.$$

We can provide a bound on the confidence sets constructed in this way. The bound contains different divergences between $P(Y|X)$ and $Q(Y|X)$.

**Theorem D.2.** *Consider the confidence set $\Gamma_Q^\alpha(X_1^n)$ defined above. Then, there is a choice of $\beta$ that achieves the confidence $1 - \alpha$ at the efficiency rate $\gamma_n$ satisfying:*

$$n\gamma_n \leq \log\left(1 + n\mathbb{E}[TV(P(\cdot|X), Q(\cdot|X))]\right) + nH(Y|X) + n\mathbb{E}[D(P(\cdot|X)\|Q(\cdot|X))] + \sqrt{n}\sigma Q^{-1}(\alpha) + O(1),$$

*assuming $\alpha \geq \rho/\sqrt{n}\sigma^3$, and:*

$$\sigma = \left(\text{Var}\left(\log Q(Y|X)\right)\right)^{1/2}$$
$$\rho = \mathbb{E}\left(\left|\log Q(Y|X) - \mu\right|^3\right).$$

*All expectations are w.r.t. data distribution $P$.*

*Proof.* First, we can use Lemma D.1 to see that:

$$\left|\Gamma_Q^\alpha(x_1^n)\right| \leq \frac{Q^n(\Gamma_Q^\alpha(x_1^n))}{\beta}.$$

Then:

$$Q(\Gamma_Q^\alpha(x_1^n)) \leq P(\Gamma_Q^\alpha(x_1^n)) + \left|Q(\Gamma_Q^\alpha(x_1^n)) - P(\Gamma_Q^\alpha(x_1^n))\right| \leq P(\Gamma_Q^\alpha(x_1^n)) + \text{TV}\left(P(\cdot|x_1^n), Q(\cdot|x_1^n),\right.$$

which implies that:

$$\mathbb{E}[\left|\Gamma_Q^\alpha(X_1^n)\right|] \leq \frac{P(\Gamma_Q^\alpha(X_1^n)) + \mathbb{E}[\text{TV}\left(P(\cdot|X_1^n), Q(\cdot|X_1^n)\right)]}{\beta}.$$

where all the expectations are w.r.t. the data distribution $P$. We can simplify the total variation distance further as follows:

$$\mathbb{E}[\text{TV}\left(P(\cdot|X_1^n), Q(\cdot|X_1^n)\right)] \leq n\mathbb{E}[\text{TV}\left(P(\cdot|X), Q(\cdot|X)\right)].$$

We can now characterize the probability of the confidence set using the central limit theorem in a similar way:

$$\mathbb{P}\left(Q(Y_1^n|X_1^n) \leq \beta\right) \leq Q\left(\frac{-1}{\sigma\sqrt{n}}\log\beta + \frac{\sqrt{n}\mu}{\sigma}\right) + \frac{\rho}{\sqrt{n}\sigma^3}.$$

The only difference is that the moments are computed for $\log Q(Y_i|X_i)$. We will come back to their computation later. First see that $\beta$ can be chosen in a way to guarantee the confidence level we are interested in:

$$\beta \leq \exp\left(n\mu - \sigma\sqrt{n}Q^{-1}(\alpha) + \frac{1}{2\pi}\frac{\rho}{\sigma^2},\right)$$

which we use to bound the expected set size

$$\mathbb{E}[\left|\Gamma_Q^\alpha(X_1^n)\right|] \leq (1 + n\mathbb{E}[\text{TV}\left(P(\cdot|X), Q(\cdot|X)\right)])\exp\left(-n\mu + \sigma\sqrt{n}Q^{-1}(\alpha) - \frac{1}{2\pi}\frac{\rho}{\sigma^2}.\right)$$

As last step we compute the moments as follows:

$$\mu = \mathbb{E}[\log Q(Y|X)] = -H(Y|X) - \mathbb{E}[D(P(\cdot|X)\|Q(\cdot|X))]$$
$$\sigma = \left(\text{Var}\left(\log Q(Y|X)\right)\right)^{1/2}$$
$$\rho = \mathbb{E}\left(\left|\log Q(Y|X) - \mu\right|^3\right),$$

$\square$

The key penalty for the distribution mismatch is the expected KL-divergence term $\mathbb{E}[D(P(\cdot|X)\|Q(\cdot|X))]$. This result also shows that one can directly try to approximate the conditional distribution if the error can be suitably controlled. For example, in Sadinle et al. (2019), the authors used k-Nearest Neighbors, local polynomial estimator and Regularized multinomial logistic regression to approximate the conditional distributions.

# E PROOFS OF SECTION 4

## E.1 PROOF OF THEOREM 4.2

We start with the first error. To start, we need to use the following definition of Jensen-Shannon divergence:

$$\text{GJS}(T_{\mathbf{X}_1}, T_{\mathbf{X}_{\text{test}}}, \alpha) = (1+\alpha)H(T_{\mathbf{X}_1 \mathbf{X}_{\text{test}}}) - \alpha H(T_{\mathbf{X}_1}) - H(T_{\mathbf{X}_{\text{test}}}) \tag{36}$$

The proof continues as follows using the properties of types reviewed in Appendix A:

$$\mathbb{P}(H_1 \notin \Gamma_n^\alpha(\mathbf{X}_1, \mathbf{X}_2, \mathbf{X}_{\text{test}}) | \mathbf{X}_{\text{test}} \sim P_1)$$

$$= \sum_{\substack{(\mathbf{x}_1, \mathbf{x}_2, \mathbf{x}_{\text{test}}) \\ \text{GJS}(T_{\mathbf{X}_1}, T_{\mathbf{X}_{\text{test}}}, \alpha) \geq \lambda}} P_1(\mathbf{X}_1) P_2(\mathbf{X}_2) P_1(\mathbf{X}_{\text{test}})$$

$$\leq \sum_{\substack{(\mathbf{x}_1, \mathbf{x}_2, \mathbf{x}_{\text{test}}) \\ \text{GJS}(T_{\mathbf{X}_1}, T_{\mathbf{X}_{\text{test}}}, \alpha) \geq \lambda}} e^{-(N+n)H(T_{\mathbf{x}_1 \mathbf{x}_{\text{test}}})} P_2(\mathbf{X}_2)$$

$$\leq \sum_{\substack{(\mathbf{x}_1, \mathbf{x}_2, \mathbf{x}_{\text{test}}) \\ \text{GJS}(T_{\mathbf{X}_1}, T_{\mathbf{X}_{\text{test}}}, \alpha) \geq \lambda}} e^{-NH(T_{\mathbf{x}_1})} P_2(\mathbf{X}_2) e^{-nH(T_{\mathbf{x}_{\text{test}}})} e^{-n\lambda}$$

$$\leq e^{-n\lambda} \sum_{\substack{(T_{\mathbf{X}_1}, T_{\mathbf{X}_{\text{test}}}) \in \mathcal{P}_N \times \mathcal{P}_n \\ \text{GJS}(T_{\mathbf{X}_1}, T_{\mathbf{X}_{\text{test}}}, \alpha) \geq \lambda}} |T_{\mathbf{X}_1}| e^{-NH(T_{\mathbf{X}_1})} |T_{\mathbf{X}_t}| P_2(\mathbf{X}_2) e^{-nH(T_{\mathbf{X}_{\text{test}}})}$$

$$\leq e^{-n\lambda} |\mathcal{P}_N \times \mathcal{P}_n| \leq e^{-n\lambda} (n+1)^{|\mathcal{X}|} (N+1)^{|\mathcal{X}|}.$$

In the last inequality, we used a bound on the number of sequences of length $N$ for each type $T$. The other error follows from a similar analysis. We have shown that the errors $\beta_1(\psi_{1,n}^{\text{Gutman}} | P_1, P_2)$ and $\beta_1(\psi_{2,n}^{\text{Gutman}} | P_1, P_2)$ are both bounded by $e^{-n\tilde{\lambda}}$, where

$$\tilde{\lambda} = \lambda - \frac{|\mathcal{X}| \log(n+1)(N+1)}{n}.$$

This implies:

$$P_e^n \leq e^{-n\tilde{\lambda}},$$

and establishes the desired result.

## E.2 PROOF OF THEOREM 4.3

First note that:

$$\mathbb{P}(|\Gamma_n^\alpha(\mathbf{X}_1, \mathbf{X}_2, \mathbf{X}_{\text{test}})| = 2) = \mathbb{P}(\text{GJS}(T_{\mathbf{X}_1}, T_{\mathbf{X}_{\text{test}}}, \alpha) < \lambda, \text{ and } \text{GJS}(T_{\mathbf{X}_2}, T_{\mathbf{X}_{\text{test}}}, \alpha) < \lambda) \tag{37}$$

Let's start as follows:

$$\mathbb{P}(|\Gamma_n^\alpha(\mathbf{X}_1, \mathbf{X}_2, \mathbf{X}_{\text{test}})| = 2) =$$
$$\pi_1 \mathbb{P}(|\Gamma_n^\alpha(\mathbf{X}_1, \mathbf{X}_2, \mathbf{X}_{\text{test}})| = 2 | \mathbf{X}_{\text{test}} \sim P_1) + \pi_2 \mathbb{P}(|\Gamma_n^\alpha(\mathbf{X}_1, \mathbf{X}_2, \mathbf{X}_{\text{test}})| = 2 | \mathbf{X}_{\text{test}} \sim P_2)$$

Let $E_1$ and $E_2$ denote respectively the events $\text{GJS}(T_{\mathbf{X}_1}, T_{\mathbf{X}_{\text{test}}}, \alpha) < \lambda$, and $\text{GJS}(T_{\mathbf{X}_2}, T_{\mathbf{X}_{\text{test}}}, \alpha) < \lambda$. Then, we have:

$$\mathbb{P}(|\Gamma_n^\alpha(\mathbf{X}_1, \mathbf{X}_2, \mathbf{X}_{\text{test}})| = 2 | \mathbf{X}_{\text{test}} \sim P_1) = \mathbb{P}(E_1 \cap E_2 | \mathbf{X}_{\text{test}} \sim P_1)$$

From the error analysis of classical Gutman's decision, we have:

$$\mathbb{P}(E_1 | \mathbf{X}_{\text{test}} \sim P_1) \geq 1 - e^{-n\tilde{\lambda}}.$$

Therefore, the probability of cardinality 2 is dominated by the event $E_2$, in the following sense:

$$\mathbb{P}(E_2 | \mathbf{X}_{\text{test}} \sim P_1) - e^{-n\tilde{\lambda}} \leq \mathbb{P}(E_1 | \mathbf{X}_{\text{test}} \sim P_1) + \mathbb{P}(E_2 | \mathbf{X}_{\text{test}} \sim P_1) - 1 \leq \mathbb{P}(E_1 \cap E_2 | \mathbf{X}_{\text{test}} \sim P_1) \leq \mathbb{P}(E_2 | \mathbf{X}_{\text{test}} \sim P_1)$$

Now using Theorem A.3, we can characterize the exponent as follows:

$$\lim_{n\to\infty}\frac{1}{n}\log\mathbb{P}(E_2|\mathbf{X}_{\text{test}}\sim P_1) = -\inf_{\substack{(Q_1,Q_2)\in\mathcal{P}^2\\ \text{GJS}(Q_1,Q_2,\alpha)<\lambda}} D(Q_2\|P_1)+\alpha D(Q_1\|P_2) = -F(P_2,P_1,\alpha,\lambda),$$

where, to remind, we had:

$$F(P_1,P_2,\alpha,\lambda) := \min_{\substack{(Q_1,Q_2)\in\mathcal{P}(\mathcal{X})^2\\ \text{GJS}(Q_1,Q_2,\alpha)<\lambda}} D(Q_2\|P_2)+\alpha D(Q_1\|P_1). \tag{38}$$

This, in turn, would imply that:

$$\lim_{n\to\infty}\frac{1}{n}\log\mathbb{P}(E_1\cap E_2|\mathbf{X}_{\text{test}}\sim P_1) \leq -F(P_2,P_1,\alpha,\lambda).$$

Next, we characterize the case $\mathbf{X}_{\text{test}}\sim P_2$, for which we similarly have:

$$\mathbb{P}(E_2|\mathbf{X}_{\text{test}}\sim P_2) \geq 1-e^{-n\tilde{\lambda}}.$$

And the even $E_1$ is nothing but the second error of classical Gutman's test:

$$\lim_{n\to\infty}\frac{1}{n}\log\mathbb{P}(E_1|\mathbf{X}_{\text{test}}\sim P_2) = -\inf_{\substack{(Q_1,Q_2)\in\mathcal{P}^2\\ \text{GJS}(Q_1,Q_2,\alpha)<\lambda}} D(Q_2\|P_2)+\alpha D(Q_1\|P_1) = -F(P_1,P_2,\alpha,\lambda),$$

Putting these results together, we obtain the following:

$$\lim_{n\to\infty}\frac{1}{n}\log\mathbb{P}(|\Gamma_n^\alpha(\mathbf{X}_1,\mathbf{X}_2,\mathbf{X}_{\text{test}})|=2) \leq -\min\left(F(P_1,P_2,\alpha,\lambda),F(P_2,P_1,\alpha,\lambda)\right). \tag{39}$$

and the equality obtains if $\min\left(F(P_1,P_2,\alpha,\lambda),F(P_2,P_1,\alpha,\lambda)\right)<\lambda$, as in the lower bound $e^{-n\tilde{\lambda}}$ vanishes faster.

The probability of having an empty set is controlled similarly, with the difference that the complement of the above events is considered.

### E.3 PROOF OF THEOREM 4.5

Let $E_i$ denote the event $\text{GJS}(T_{\mathbf{X}_i},T_{\mathbf{X}_{\text{test}}},\alpha)<\lambda$. We first condition on the event that $\mathbf{X}_{\text{test}}$ follows the distribution $P_l$. The key event is the following:

$$\mathbb{P}(|\Gamma_n^\alpha(\mathbf{X}_1,\ldots,\mathbf{X}_m,\mathbf{X}_{\text{test}})|=k|\mathbf{X}_{\text{test}}\sim P_l) = \mathbb{P}\left(\bigcup_{S\subset[M]:|S|=k}\left(\bigcap_{i\in S}E_i\right)\bigcap\left(\bigcap_{i\in S^c}E_i^c\right)\Bigg|\mathbf{X}_{\text{test}}\sim P_l\right)$$

We will use the union bound, and therefore focus on the following probabilities for $l\in S$ and $l\notin S$. We start with $l\in S$, for which we get:

$$\mathbb{P}\left(\left(\bigcap_{i\in S}E_i\right)\bigcap\left(\bigcap_{i\in S^c}E_i^c\right)\Bigg|\mathbf{X}_{\text{test}}\sim P_l\right) \leq \mathbb{P}\left(\left(\bigcap_{i\in S,i\neq l}E_i\right)\Bigg|\mathbf{X}_{\text{test}}\sim P_l\right), \tag{40}$$

where the removed events in the upper bound have all probabilities converging to 1. The latter probability decays exponentially fast with the exponent following from Sanov's theorem:

$$\lim_{n\to\infty}\frac{1}{n}\log\mathbb{P}\left(\left(\bigcap_{i\in S,i\neq l}E_i\right)\Bigg|\mathbf{X}_{\text{test}}\sim P_l\right) \leq -\inf_{\substack{(Q_1,\ldots,Q_M,Q_t)\in\mathcal{P}^{M+1}\\ \text{GJS}(Q_i,Q_t,\alpha)<\lambda,\forall i\in S/\{l\}}} \alpha\sum_{i=1}^{M}D(Q_i\|P_i)+D(Q_t\|P_l)$$

$$= -\inf_{\substack{((Q_i)_{i\in S/\{l\}},Q_t)\in\mathcal{P}^{|S|}\\ \text{GJS}(Q_i,Q_t,\alpha)<\lambda,\forall i\in S/\{l\}}} \alpha\sum_{i\in S/\{l\}}D(Q_i\|P_i)+D(Q_t\|P_l)$$

Similarly, for $l\notin S$, we have:

$$\mathbb{P}\left(\left(\bigcap_{i\in S}E_i\right)\bigcap\left(\bigcap_{i\in S^c}E_i^c\right)\Bigg|\mathbf{X}_{\text{test}}\sim P_l\right) \leq \mathbb{P}\left(\left(\bigcap_{i\in S}E_i\right)\bigcap E_l^c\Bigg|\mathbf{X}_{\text{test}}\sim P_l\right), \tag{41}$$

which leads to the following exponent:

$$\lim_{n\to\infty} \frac{1}{n} \log \mathbb{P}\left(\left(\bigcap_{i\in S} E_i\right) \bigcap E_l^c \middle| \mathbf{X}_{\text{test}} \sim P_l\right) \leq - \inf_{\substack{(Q_1,\ldots,Q_M,Q_t)\in\mathcal{P}^{M+1} \\ \mathrm{GJS}(Q_i,Q_t,\alpha)<\lambda,\forall i\in S \\ \mathrm{GJS}(Q_l,Q_t,\alpha)\geq\lambda}} \alpha \sum_{i=1}^{M} D(Q_i\|P_i) + D(Q_t\|P_l)$$

$$= - \inf_{\substack{(Q_1,\ldots,Q_M,Q_t)\in\mathcal{P}^{M+1} \\ \mathrm{GJS}(Q_i,Q_t,\alpha)<\lambda,\forall i\in S \\ \mathrm{GJS}(Q_l,Q_t,\alpha)\geq\lambda}} \alpha \sum_{i\in S\cup\{l\}} D(Q_i\|P_i) + D(Q_t\|P_l)$$

So using the definition of $F(\{P_i : i \in S\}, P_l, \alpha, \lambda)$, we get:

$$\lim_{n\to\infty} \frac{1}{n} \log \mathbb{P}(|\Gamma_n^\alpha(\mathbf{X}_1,\ldots,\mathbf{X}_m,\mathbf{X}_{\text{test}})| = k | \mathbf{X}_{\text{test}} \sim P_l) \leq - \inf_{S\subset[M],|S|=k} F(\{P_i : i \in S\}, P_l, \alpha, \lambda)$$

(42)

which, in turn, implies the final result by finding the smallest exponent for each $l$.

For the probability of the empty prediction set, we use a simple upper bound that the true label does not belong to the prediction set. This probability decays exponentially with the exponent $-\lambda$.

# F   DISCUSSION ON OPTIMALITY AND SECOND ORDER ANALYSIS FOR EMPIRICALLY OBSERVED STATISTICS

**Discussion on Optimality.**   We stated in the paper that the classical Gutman's test was optimal (Zhou et al., 2020). To repeat, for any other decision rule $\phi_n$ that *uniformly* controls the error exponent of $\beta_1(\psi_n|P_1, P_2)$ similar to Gutman, namely

$$\forall(P_1, P_2) \in \mathcal{P}(\mathcal{X})^2 : \liminf_{n\to\infty} -\frac{1}{n} \log \beta_1(\psi_n|P_1, P_2) \geq \lambda,$$

then, the type-II error is always worse than Gutman's test:

$$\beta_2(\psi_n|P_1, P_2) \geq \beta_2(\psi_n^{\text{Gutman}}|P_1, P_2).$$

We can use this optimality result to provide a heuristic argument for the efficiency of Gutman's test with confidence. Consider all the confidence predictors that satisfy similar error exponents for $\beta_1(\psi_n|P_1, P_2)$. Given the optimality of Gutman's classical test, the type-II error is higher, which also means that the probability of having an undesirable term in the set increases. In other words, the inefficiency of the test increases.

**Second Order Analysis.**   For non-asymptotic results, the following limit is controlled in (Zhou et al., 2020) for classical tests (no confidence predictor):

$$\lambda(n, \alpha, \epsilon, \mathbf{P}) := \sup\left\{\lambda \in \mathbb{R}_+ : \exists\psi_n \text{ s.t. } \forall j \in [2], \forall(\tilde{P}_1, \tilde{P}_2) \in \mathcal{P}(\mathcal{X})^2 :\right.$$

$$\left.\beta_1(\psi_n|P_1, P_2) \leq \exp(-n\lambda); \beta_2(\psi_n|P_1, P_2) \leq \epsilon\right\}.$$

This definition is for binary hypothesis tests, but it can be extended further to multiple hypothesis test. The following result characterizes the limit.

**Theorem F.1** (Theorem 2 (Zhou et al., 2020)). *For any $\epsilon \in (0, 1)$, and any distribution $(P_1, P_2) \in \mathcal{P}(\mathcal{X})^2$, the second order limit is characterized as follows:*

$$\lambda(n, \alpha, \epsilon, \mathbf{P}) = \mathrm{GJS}(P_1, P_2, \alpha) + \sqrt{\frac{V(P_1, P_2, \alpha)}{n}}\Phi^{-1}(\epsilon) + O\left(\frac{\log n}{n}\right),$$

*where $\Phi(t) = 1 - \mathrm{Q}(t)$ is the cumulative distribution function of the standard normal Gaussian distribution, and the dispersion function is defined as*

$$V(P_1, P_2, \alpha) = \alpha\mathrm{Var}_{P_1}\left(\log \frac{(1+\alpha)P_1(X)}{\alpha P_1(X) + P_2(X)}\right) + \mathrm{Var}_{P_2}\left(\log \frac{(1+\alpha)P_2(X)}{\alpha P_1(X) + P_2(X)}\right).$$

Note that this result controls the rate of the second error, which is about yielding the wrong hypothesis, while maximizing the decay rate of the first error. The dual setting of this problem where the first error is controlled, as desired in our setup, is also studied in (Zhou et al., 2020), Proposition 4. However, the result is still for asymptotic $n$, as $n$ goes to infinity. As one can expect, the asymptotic limit will be based on the error exponent $F(P_1, P_2, \alpha, \lambda)$ where $\lambda \to 0$. This limit turns out to be the Rényi divergence of order $\alpha/(1 + \alpha)$. In any case, such result is not useful in our context.

We can use still use the result of Theorem F.1 to build a confidence predictor. We provide a high level idea of such construction. The proofs can be formalized in a similar way to the other proofs. We combine two tests, one that controls $\beta_2(\psi_{1,n}|P_1, P_2) \leq \epsilon$ and the other controlling $\beta_1(\psi_{2,n}|P_1, P_2) \leq \epsilon$. We use $\psi_{1,n}$ only to decide on the inclusion of $H_1$ and $\psi_{2,n}$ on the inclusion of $H_2$. This is a similar procedure to Definition 4.1. Since the second errors are controlled, we can immediately see that

$$\mathbb{P}(|\Gamma_n^\alpha(\mathbf{X}_1, \mathbf{X}_2, \mathbf{X}_{\text{test}})| = 2) \leq \epsilon.$$

Therefore the efficiency can be effectively controlled in this fashion. However, the confidence cannot then be arbitrarily controlled. Two exponents control the confidence error $\text{GJS}(P_1, P_2, \alpha) + \sqrt{\frac{V(P_1, P_2, \alpha)}{n}} \Phi^{-1}(\epsilon)$ and $\text{GJS}(P_2, P_1, \alpha) + \sqrt{\frac{V(P_2, P_1, \alpha)}{n}} \Phi^{-1}(\epsilon)$. The best exponent will be the minimum of both. Deriving non-asymptotic bounds for the dual setting can be an interesting future work.

## G    COMPARISON WITH PRIOR WORKS

**Comparison with (Correia et al., 2024).**    The authors in (Correia et al., 2024) provided information theoretic bounds on the efficiency of conformal prediction algorithms. The main bound on the expected set size is derived from Fano's inequality for variable size list decoding, given in Proposition C.7 of the paper:

$$H(Y|X) \leq h_b(\alpha) + \alpha \log |\mathcal{Y}| + \mathbb{E}([\log |\mathcal{C}(x)|]^+),$$

This is for a single test sample prediction. Since the bound holds for any space $\mathcal{Y}$ and $\mathcal{X}$, we can use it for transductive confidence prediction by choosing the product space $\mathcal{Y}^n$ and $\mathcal{X}^n$, which yields the following bound, assuming independent samples:

$$nH(Y|X) \leq h_b(\alpha) + n\alpha \log |\mathcal{Y}| + n\mathbb{E}([\log |\mathcal{C}(X)|]^+),$$

As $n \to \infty$, and using Jensen's inequality, we get:

$$H(Y|X) \leq \alpha \log |\mathcal{Y}| + \gamma_m^-.$$

The result implies that if $\gamma_m^- < H(Y|X)$, then

$$\alpha \geq \frac{H(Y|X) - \gamma_m^-}{\log |\mathcal{Y}|},$$

which means that the value $\alpha$ cannot be made arbitrarily small (or confidence arbitrarily high). Our result, as stated in Theorem 3.2 is stronger, as it says that in such case the confidence goes to zero, or $\alpha \to 0$.

This is analogous to the results in information theory about weak and strong converses for Shannon capacity. The weak converse is proven using Fano's inequality and states that the rates above the capacity cannot have zero error. The strong converse states that the error goes to one. Fano's inequality is known to be loose in certain scenarios, which motivated many works on more efficient and tighter bounds in information theory (see (Polyanskiy et al., 2010) and references therein).

**Transductive conformal prediction in (Vovk, 2013).**    As shown in (Vovk, 2013), it should be noted that transductive conformal predictors are a class of transductive confidence predictors. Our theoretical bounds apply to all confidence predictors, which constitute a larger class. However, Theorem 3 in (Vovk, 2013) states, there is always a conformal predictor as good as a transductive one. Therefore, throughout the paper, we used mainly conformal predictors as our focus. However, the notion of nonconformity score, essential for transductive prediction, was not discussed in the paper. We review the confidence predictor using nonconformity score.

Transductive conformal predictor as in (Vovk, 2013) is defined using a *transductive nonconformity score* $A : (\mathcal{X} \times \mathcal{Y})^* \times (\mathcal{X} \times \mathcal{Y})^* \to \mathbb{R}$ where $(\mathcal{X} \times \mathcal{Y})^*$ is the set of all finite sequence with elements $(X, Y)$, $X \in \mathcal{X}, Y \in \mathcal{Y}$. $A(\zeta_1, \zeta_2)$ does not depend on the ordering of $\zeta_1$. The transductive conformal predictor for $A$, based on the labeled dataset given as $Z_1^m = ((X_i, Y_i) : i \in [m])$, compute the transductive nonconformity scores for each possible labels $\boldsymbol{v} = (v_{m+1}, \ldots, v_{m+n}) \in \mathcal{Y}^n$ of the test sequence $X_{m+1}^{m+n} = (X_{m+1}, \ldots, X_{m+n})$ as follows. Construct the labels $Y_{m+k}^{\boldsymbol{v}} = v_{m+k}$ for $k \in [n]$, $Y_i^{\boldsymbol{v}} = Y_i$ for $i \in [m]$. Consider the following definition:

$$\boldsymbol{Z}_S^{\boldsymbol{v}} = ((X_i, Y_i^{\boldsymbol{v}}) : i \in S).$$

Then, for each possible labels $\boldsymbol{v} = (v_{m+1}, \ldots, v_{m+n}) \in \mathcal{Y}^n$ and each ordered subset $S$ of $[m+n]$ with $n$ entries define:

$$\xi_S^{\boldsymbol{v}} := A(\boldsymbol{Z}_{[m+n] \setminus S}^{\boldsymbol{v}} \boldsymbol{Z}_S^{\boldsymbol{v}}). \tag{43}$$

and use to compute $p$-values:

$$p(v_1, \ldots, v_n) = \frac{|S : \xi_S^{\boldsymbol{v}} \geq \xi_{\boldsymbol{v}}^{\boldsymbol{v}}|}{(m+n)!/n!}.$$

These $p$-values can be used to construct the prediction sets as follows:

$$\Gamma^\alpha(Z_1^m, X_{m+1}^{m+n}) = \{\boldsymbol{v} = (v_{m+1}, \ldots, v_{m+n}) \in \mathcal{Y}^n : p(v_1, \ldots, v_n) \geq \alpha\}.$$

Such construction comes with theoretical coverage guarantee that the predictor has the confidence at least $1 - \alpha$ in the online mode (see Theorem 1 and Corollary 1 of (Vovk, 2013) for further discussions).

As it can be seen from the above construction, computing all these $p$-values is computationally cumbersome. Therefore, one can try to construct transductive nonconformity measures from single nonconformity measures using another aggregator. Bonferroni predictors compute $p$-value for each test sample separately and the combine that using the Bonferroni equation:

$$p := \min(np_1, \ldots, np_n, 1),$$

which amounts to the following modified prediction set:

$$\Gamma^\alpha(Z_1^m, X_{m+1}^{m+n}) = \prod_{i=1}^n \left\{ v_{m+i} \in \mathcal{Y} : p(v_{m+i}) \geq \frac{\alpha}{n} \right\}.$$

Bonferroni predictors have similar coverage guarantees to transductive conformal prediction (see Theorem 2 in (Vovk, 2013)).

For our experiments, we use Bonferroni predictors for the $p$-values obtained from split conformal prediction (SCP). Although the method works based on computing $(1 - \alpha)$-quantile, there is a 1-1 mapping to a $p$-value:

$$s \leq \text{Quantile}(1 - \alpha; \{S_i\}_{i=1}^n \cup \{\infty\}) \iff \frac{1}{n} \sum_{i=1}^n \mathbf{1}(S_i \geq s) > \alpha.$$

In other words, the term $\frac{1}{n} \sum_{i=1}^n \mathbf{1}(S_i \geq s)$ is a $p$-value. Therefore, Bonferroni predictor for SCP can be obtained by running SCP per test sample using $1 - \frac{\alpha}{n}$-quantile and then get the set product of predicted sets.

**On Optimal Confidence Predictors.** The prior works considered the efficiency-confidence trade-off in context of confidence prediction (Lei, 2014; Lei & Wasserman, 2014; Lei et al., 2013; 2015; Sadinle et al., 2019). In (Lei, 2014), the case of binary classification is discussed where the prediction sets are constructed based on the thresholding of the conditional probability. Using Neyman-Pearson lemma, it is shown that such prediction sets achieve optimal efficiency. When the conditional probability is not given, its empirical version is used, which is shown to asymptotically achieve the optimal confidence prediction. Their analysis excludes the empty prediction sets. The optimal classifier of multi-class classifiers was discussed in (Sadinle et al., 2019). The solution is similarly based on the thresholding of the conditional probability. Our work is the general derivation of lower and upper bound on the optimal prediction sets in transductive setting, and is connected to information theoretic quantities as well.

# H    SUPPLEMENTARY EXPERIMENTAL RESULTS

In this section, we present additional numerical results related to our theoretical bound. All experiments are with $N = 10$ (corresponding to MNIST), and follows a similar setup presented in the main paper.

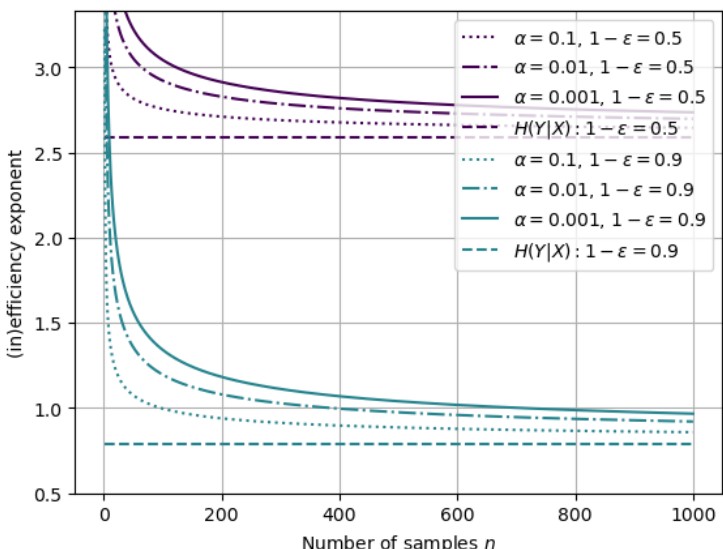

**Figure 2:** The theoretical finite block length bounds for different noise levels, and different confidence $\alpha$

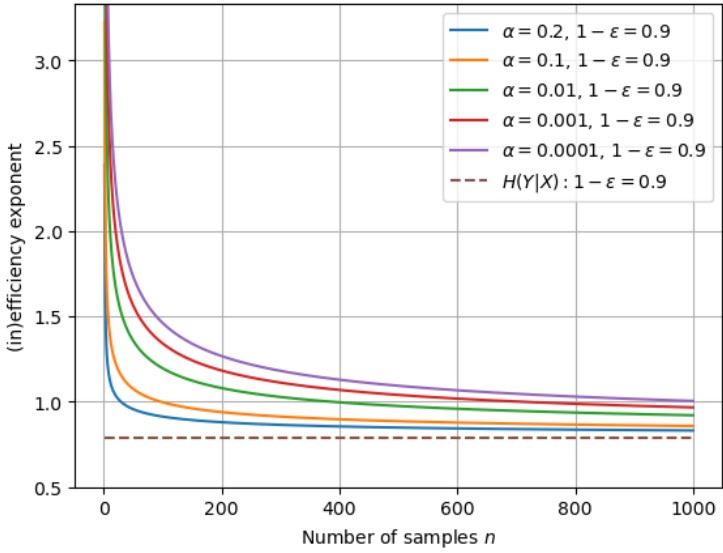

**Figure 3:** The theoretical finite block length bounds for different confidence $\alpha$ in terms of $n$

**Simulating Theoretical Bounds.**    Figures 2, 3, and 4 are all based on simulating the theoretical bounds for noisy labels given in eq. 5. We would like to observe a few trends, most of them intuitively expected. In Figure 2 and 3, we plot the finite block-length bounds as a function of the number of test samples $n$ for different level of confidence. As the level of required confidence becomes more stringent, namely smaller $\alpha$, the inefficiency, given by the exponent of the expected set size, increases. Besides, the finite block length bound approaches slowly toward the asymp-

totic bound, $H(Y|X)$. Figure 2 plots the bounds for two different noise levels, which shows that changing noise level, i.e. intrinsic uncertainty, has a more drastic impact on the inefficiency.

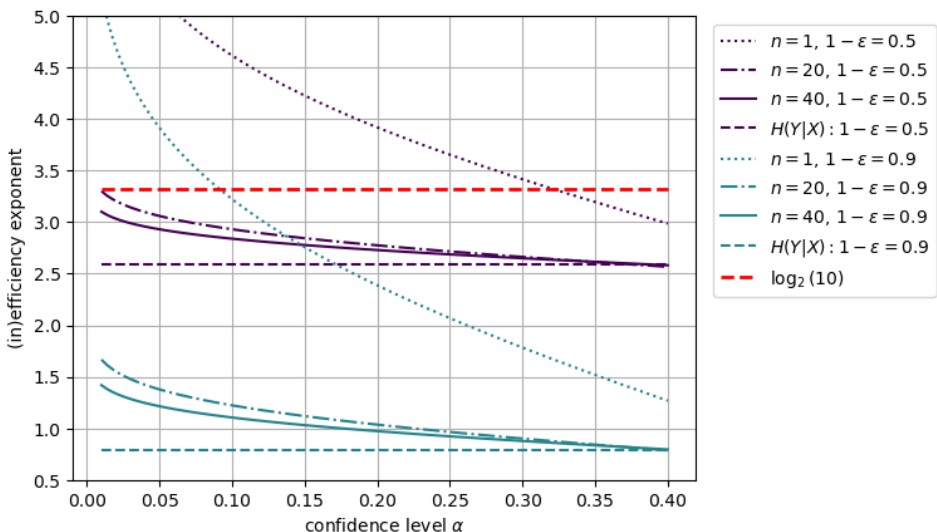

**Figure 4:** The theoretical finite block length bounds for different number of test samples $n$ in terms of confidence $\alpha$

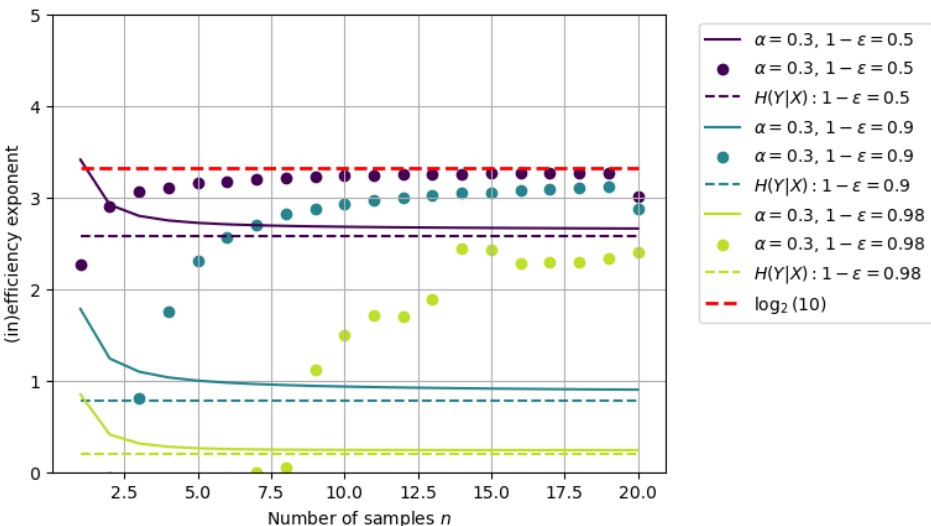

**Figure 5:** The comparison of the upper bound with naive Bonferroni split conformal prediction for transductive inference - $\alpha = 0.3$

In Figure 4, we plot the bounds in terms of confidence levels. As we decrease the required confidence level by choosing larger $\alpha$, the inefficiency decreases as well. Similar to previous plots, choosing smaller $n$ increases the bound.

If the full set $\mathcal{Y}$ is chosen every time as predicted set, it trivially included the correct label, but it yields the most inefficient prediction with the expected set size $\log |\mathcal{Y}|$. We plot it using the dashed red line, which shows $\log_2(10)$ for our experiment. The bound becomes vacuous, whenever it is above that line. There are a few reasons behind the vacuity of our bound. First of all, certain terms were ignored in the approximate bound, this includes $\log \Delta$, as well as a condition on $\alpha + \frac{\rho}{\sqrt{n}\sigma^3} + \Delta$ being within the interval $(0, 1)$. We discuss these details in Remark B.3.

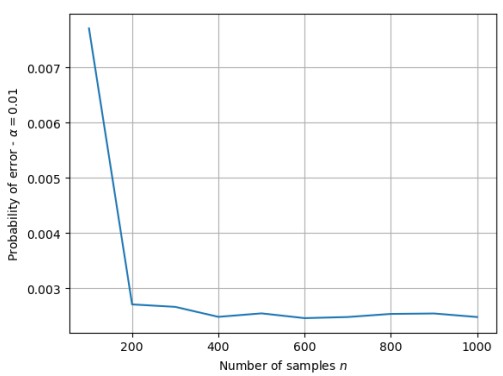
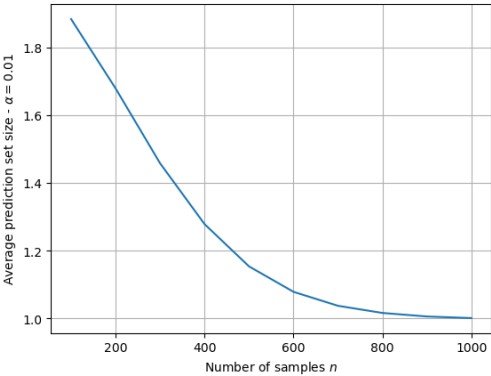

**(a)** The error probability of the test as a function of the data size

**(b)** The average prediction set size as a function of the data size

**Figure 6:** Numerical Results for the confidence based Gutman's test

**Numerical Comparisons with Transductive Methods.** We provide another plot for the Bonferroni transductive method in Figure 5 for two different levels of confidence $\alpha = 0.1$ and $\alpha = 0.3$. Bonferroni predictors in (Vovk, 2013) were discussed in Section G. The idea is to convert per-sample $p$-values to $p$-value for transductive prediction. To have a transductive prediction of level $\alpha$ for $n$ samples, we find predictions sets for each sample at the level $\alpha/n$, and compute the set product. Our experiment setup remains the same with 180 samples in the calibration set. We observe a similar inefficiency of Bonferroni prediction as $n$ increases. Besides, as explained, the approximate bound can be loose for smaller $n$. In particular, for noisier datasets, the bound takes longer to be non-vacuous. Another discrepancy between the bound and the experiment is that the bound assumed full knowledge of the conditional distribution $\mathbb{P}(Y|X)$. However, in our experiments, we only have access to the samples. This is expected to incur an additional gap with the bound. Nonetheless, these experiments still provide a better bound than $H(Y|X)$ as reported in (Correia et al., 2024), and highlight the room for improvement in the transductive methods.

**Numerical Simulation of Confidence Gutman's test.** We conduct experiments using the same setup as Zhou et al. (2020). We have chosen the binary classification task where two classes correspond to Bernoulli distributed sequences with parameters 0.2 and 0.6. We have selected the ration of training to test samplpes as 0.2, namely $N = 0.2n$. The number of test samples are chosen as $\{100, 200, \ldots, 1000\}$. We selected the confidence level 0.01. The experiments are averaged over $10^6$ runs. The results can be seen in Figure 6 where we plotted the average prediction set size and the error probability as a function of the dataset size. As predicted by the theory, the proposed test quickly coverges to the average prediction set size of one, while the error probability lies below the prescribed confidence level 0.01, and will go down as well with $n$.

