# OpenReview forum: "Fundamental bounds on efficiency-confidence trade-off for transductive conformal prediction"
_ICLR.cc/2026/Conference — Submitted to ICLR 2026_

### Official Review · Reviewer_psdB · 2025-10-15

**Soundness:** 3
**Presentation:** 3
**Contribution:** 2
**Rating:** 4
**Confidence:** 5

**Summary:**

This submission is essentially split into two separate parts: first, it provides bounds on the set size for "transductive" CP by assuming a known data distribution; and then it provides a set-based extension of a known result by Gutman on a special case of the problem.

The first part builds on standard information-theoretic tools, while the second leverage standard results on binary hypothesis testing with training data.

Experimental results are provided for the first part only.

**Strengths:**

The two parts of the submission, while limited on their own, offer a useful reference for researchers interested in "transductive" CP (defined here as the problem of producing a set prediction for a batch of test samples).

The paper is clearly and formally written, and useful pointers are provided to the literature.

**Weaknesses:**

The analysis in Section 3 essentially assumes knowledge of the data distribution. While it is true that this assumption yields upper  bounds on the true performance of transductive CP, it is also the case that the assumption appears to completely hide the role of the calibration data size m.

Furthermore, the results of the analysis appear to be rather expected and limited in scope. The typical set of a sequence of i.i.d. variables grows exponentially with the entropy, and so must also any prediction set with non-vanishing coverage. The result in Theorem 3.4 is also a refinement of the same idea.

The setting studied in Section 4 is a direct extension of Gutman's work on  binary hypothesis testing with training data. In fact, most of the section is devoted to reviewing existing results, and the new contribution follows directly by reframing the problem as one of set prediction.

No experimental results are provided for the setting studied in Section 4.

**Questions:**

1) How can the analysis in Section 3 be extended to provide insights on the role of the size of the calibration dataset?

2) Can Section 4 be rewritten to address directly the case with any number of hypotheses?

3) If so, can Theorem 5 be simplified to provide clearer insights into the average prediction set size?

4) How do the results in Section 4 connect to the theorem in Section 3?

5) Can experimental results be provided to relate the material in Section 3 and Section 4?

---

> ### Author Response · Authors · 2025-12-03
> **Response to the reviewer**
>
> We would like to thank the reviewer for the careful reading of the paper and relevant questions and feedback.
>
> * **On assuming the  knowledge of the data distribution.** We would like to highlight that although the data distribution plays a key role in the lower bound on the prediction set size, the practical methods based on the calibration data can still benefit from the insights. For example, in case on classical conformal prediction, the conditional probability is not given, and its empirical version is used, which is shown to asymptotically achieve the optimal confidence prediction - see for example [1]. There are many works along the same line. We added a new paragraph at the end of Appendix G.
>
> * **On the novelty of the result.** We would like to highlight that our theoretical results are a bit more nuanced. In general, in information theory, there is this common intuition that asymptotic i.i.d. random events show a form of typicality. The main challenge in the field to precise characterize the trade-offs, for example between the compression rate and entropy, between achievable rate regions and mutual information. There are many instances where the exact characterization of these trade-offs are still open. In our case, we precisely characterize the relation of entropy, expected prediction set size and the confidence level. These derivations require  new inequalities as in Theorem 3.1. Theorem 3.4 involves similar challenges. Note that despite many works on second order rates for instance by Strassen in 1962, the precise characterization of those rates happened only in 2010, showing the challenge of such works.
>
> * **On Gutman's result.** We decided to start with reviewing Gutman's results, since from our experience, the general audience is less familiar with this framework, and it is easier to grasp if we start with simpler and smaller examples. Even here, the extension of Gutman's result to transductive setting requires the intuition that the reject hypothesis is no longer needed, and besides, the GJS divergence are monitored individually. This is very different from classical Gutman's result where the divergences are considered jointly to decide on the hypothesis - see Theorem 2 of [2].
>
> * **Insights on the size of the calibration dataset.** This is an interesting question. If the calibration dataset is used to approximate the conditional probability distribution as in [1], then we can include the approximation error in the bound. We have added Theorem D.2 which provides a bound on the prediction set size in terms of information theoretic quantities as well as the approximation error. The data can be used to approximate the conditional distribution, and the approximation error can be bounded using concentration inequalities as a function the calibration set size.
>
> * **Regarding Section 4 and Theorem 5.** We can re-write  Section 4 to address directly the case with any number of hypotheses; similarly, we can do the same for Theorem 5 to directly state it in terms of the average prediction set size. We would like to provide our reasoning behind the current choice of our flow. Given the complexity of the results in Section 4 for general audience, we decided to build up the result gradually from simpler cases. For instance compare Theorem 4.5 with Theorem 4.3. Similarly, the direction presentation of the averaged prediction set size can lead to a complex expression that lacks intuition about the dominant terms. Therefore, we decided to decompose the results into simpler and more intuitive fragments.
>
>
> * **Connection between Section 4 and Section 3.** The bounds in Section 3 are all obtained for general transductive setting where the test samples have different labels. In Section 4, we look at the special case where the labels are identical. Therefore, the bounds in Section 3 will be naturally loose when applied to Section 4. To expand further, suppose that all the classes are equiprobable. In this case, the conditional distribution $P(Y=k|X)$ is equal to $P_k/\sum_{i=1}^M P_i$. This means that the conditional entropy is in general non-zero and therefore the averaged prediction set size grows exponentially. However, when the test labels are assumed to be identical, the results of Section 4 shows that the prediction set size can converge to 1 asymptotically.
>
> * **Experiments for Section 4.** We have added a new experiment related to Section 4.
>
> [1] Mauricio Sadinle, Jing Lei, and Larry Wasserman. Least ambiguous set-valued classifiers with
> bounded error levels. Journal of the American Statistical Association, 114(525):223–234, 2019.
>
> [2] Michael Gutman. Asymptotically optimal classification for multiple tests with empirically observed statistics. IEEE Transactions on Information Theory, 35(2):401–408, 1989.

---

### Official Review · Reviewer_MyY1 · 2025-10-30

**Soundness:** 3
**Presentation:** 1
**Contribution:** 3
**Rating:** 8
**Confidence:** 2

**Summary:**

In standard setup for conformal prediction we guarantee the coverage per each test point. This paper addresses the problem for joint prediction sets where the guarantee is to be derived for joint coverage — all labels should be in all prediction sets within a batch to be count as a coverage. The authors pose two questions, one on the information theoretic bound over the trade off between the set size and the joint coverage, and the other is how can er find an optimal way to leverage the entire labeled set (including the training set) for conformal prediction without sacrificing the coverage guarantee (conventionally, this guarantee breaks if we introduce training points to the process).

The authors provide lower bound on this setup showing that either the set size increases drastically or the guarantee vanishes to zero by increasing the number of points over which the decision is jointly made.

Furthermore they discuss a specific case where the joint prediction is over a single label -- the case where a single datapoint is examined several times for robustness.

*Disclaimer.* I should note that due to the representation issues I mentioned in the weaknesses, I could not fully follow the paper.

**Strengths:**

1. The targetted problem is very interesting (however in the naming it intersects with full conformal prediction). Being able to provide joint guarantee better than bounds from the Bonferroni correction is applicable.
2. The theory of the paper is concretely written, and (to the best of my understanding during reading the paper) all theorems are proved solidly.
3. The connection of their work to robustness (while it should not be mistaken with the adversarial and probabilistic robustness) is very interesting.

**Weaknesses:**

**Introduction is not easy to read.** Surely the introduction is written with a solid mathematical notation and there is no issues with that. But it is at least not friendly to non-expert reader. I would suggest elaborating more to the comparison of inductive and transductive conformal prediction for example and offer a one sentence brief explanation by what you mean when introducing a new concept. One really helpful way to improve the readability is to directly say that “by transductive conformal inference on a batch of datapoints we are interested on he probability that all labels are within all prediction sets”. I know that formally it can be inferred from the text, but explicitly saying that in the introduction increases the reading speed considerably.

The authors introduce two interesting questions in Section 2, while I can not see a footprint of those questions (specifically the second one) in the introduction.

Even the setup with all equal labels is not presented as it is in the introduction. At least it could be better if the authors mentioned the robust prediction application when introducing the setup for the first time.

**General readability.** The paper (due to the subject) is already not easy to read, and the authors sometimes introduce a new notation without fixing it. For instance $P_e$ in line 48. The authors also do not provide any intuitive message from the theorems they proof (for example in theorem 3.1 I can not parse what the theorem is trying to say about the joint probabilities).

**Limited Experimental Setup.** The paper only introduces numerical results on MNIST dataset. I do not count this as a negative point in my score as the paper is a theory paper. The question remains that how their results could be written in terms of a lower bound for any dataset. If the authors provide a clear algorithmic approach to derive the bounds over any dataset, then I would increase my score.

**Questions:**

1. In line 54 is the term $P(Y^{m+n}_{m+1}m | X^{m+n}_{m+1}m)$ equal to the product of all conditional probabilities from m+1 to m+n? If so, can you elaborate why it not a function of the number of elements? From reading your proof I assume this is because the value alpha already encodes the number of the elements but I am not sure why.
2. Is the entropy mentioned in Theorem 3.2, written in terms of the true confidence? Is there any bounds on the accuracy of the model?
3. What is delta in theorem 3.4? Also is there any intuitive understanding about the other variables sigma and rho? What are they encoding?
4. How your results can be expressed in terms of a guaranteed upper bound on the joint guarantee for any dataset? Is this bound affected by the number of classes or the quality of the model? Or is there any need to exactly specify the ground truth p(y|x) to derive these bounds?

---

> ### Author Response · Authors · 2025-12-03
> **Response to the reviewer**
>
> We would like to thank the reviewer for the careful reading of the paper and fruitful comments and feedback.
>
> * **Regarding the presentation.** We understand that the paper can be demanding to read for non-expert readers. However, we have tried our best to provide intuitions and introduce the concepts gradually. We will keep the reviewer's comment in mind while finalizing the manuscript and try to add more intuitions and clarity.
>
> * **Regarding the introduction.** The reviewer mentioned that some questions mentioned in the paper are not motivated in the introduction. However, we have included explicit bullet points in the introduction that matches exactly the content of the following sections. We would like to kindly ask the reviewer to clarify further their comments.
>
> * **Regarding the applicability of the bound for any dataset.** We would like to highlight that the entropy bound is general and applicable to any scenarios. There are cases for which the  labels are noisy and the noise model are known in advance. In this case, we can compute the entropy terms. However, to goal of our framework is not to approximate the entropy term using the data. On the contrary, we propose to use our theoretical bounds in synthetic settings, where the conditional probability is known, and the bounds are computable. Then, we can evaluate how close the existing transductive methods are to the fundamental bound. This is in analogy with research in field of information theory where binary symmetric channels, erasure channels and AWGN channels are used as benchmark for designing error correction codes. We use this vision in our numerical result section, where we evaluated the Bonferroni method and showed its shortcoming. We believe our theory opens up a research program on design and evaluation of transductive prediction methods under different noise regimes.
>
> * **Regarding the term $P(Y^{m+n}{m+1} | X^{m+n}{m+1})$ and the number of elements:**  We assume that the reviewer is referring to the fact that our bound does not depend on the number of training (or calibration) samples $m$. Our bound is a fundamental limit on the performance of any transductive confidence predictor, even those predictors with access to the underlying conditional distribution, which are called ideal predictors in context of conformal prediction - see Chapter 4 of [1]. As we have shown, the optimal predictors can be constructed with the knowledge of the conditional distribution $P(Y|X)$ and there is no additional gain in using calibration data. However, if we use data to approximate the conditional distribution, the number of samples impact the approximation error. We have introduced Theorem D.2 to address this case.
>
> * *Entropy term and confidence.** The entropy term in Theorem 3.2 is the property of data distribution. Therefore, the model does not play any role in it. The confidence level is also fixed in advance and is independent of the model. We have discussed more on the role of the accuracy of the model in other answers.
>
> * **Intuition about the terms $\Delta, \sigma, \rho$.** The term $\Delta$ is a free parameter that can be chosen subject to the conditions specified in Theorem 3.4. Note that the conditional entropy is the expected value of negative log conditional probability. The term $\sigma$ and $\rho$ are the variance and the centered third moment of the log conditional probability.
>
> * **The role of model and number of classes in the bound.** Our bounds are independent of the choice of the model. They show the fundamental limits of any transductive learning algorithms. However, if we use a model that approximates the conditional distribution with a certain error, the efficiency rate will be impacted proportional to the KL-divergence between the ground truth conditional distribution and the approximation. We have added Theorem D.2 that characterizes this error. To summarize, one can use the model probabilities to construct the prediction sets, and the efficiency depends on the model quality.
>
> [1] Vladimir Vovk, Alexander Gammerman, and Glenn Shafer. Algorithmic Learning in a Random
> World. Springer International Publishing, 2022.

---

### Official Review · Reviewer_FpE3 · 2025-11-01

**Soundness:** 3
**Presentation:** 2
**Contribution:** 2
**Rating:** 4
**Confidence:** 2

**Summary:**

This is a theoretical paper that characterizes the trade off between coverage and set size of transductive conformal prediction in the classification setting. In transductive conformal prediction, the goal is to construct prediction sets for $n$ test points such that all test labels fall within the joint prediction set with probability at least $1-\alpha$. The paper proves both asymptotic limits and non-asymptotic bounds for the “efficiency rate”, a normalized measure of the size of the transductive prediction set.

**Strengths:**

The paper makes some interesting connections between conformal prediction and information theory and applies some interesting tools.

**Weaknesses:**

As someone who does not have a strong background in information theory or familiarity with Gutman’s test, I found it hard to follow section 4.

**Questions:**

1. Theorem 3.1: Can you add the interpretation of this result in words? This applies to all theorem statements.
2. Do you view this work as more than simply “theory for the sake of theory”? Can this theory eventually lead to work that will inform practice?
3. On line 53, you write “When all test points share the same label, a scenario relevant to safety-critical applications…” — what is an example?

Typos/stylistic comments:
* There are multiple places where \citep is used where \citet should be used instead
* I would mention somewhere that what you call “confidence” is commonly called “coverage” in the conformal prediction literature
* Line 53: “same label” -> “same unknown label”
* Line 143: Capitalize “in”
* Line 174-175: asymptotically is used twice in this sentence
* Line 175: “In case” -> “In the case”
* Line 245: remove “setup”
* Line 249-250: I would replace the “=“ with “:=“
* Line 281: “prediction sets a single” -> “prediction sets with a single”
* Line 283: math mode error
* Theorem 4.4: “M class”? Should this be “M-class setting”?

---

> ### Author Response · Authors · 2025-12-03
> **Response to the reviewer**
>
> We would like to thank the reviewer for the careful reading of the paper and fruitful comments and feedback. We have also included all the editorial suggestions in the new version.
>
> * **Regarding the interpretation of Theorem 3.1.** Note that the prediction sets $\Gamma^\alpha$ is supposed to include $(1-\alpha)$-mass of labels.
> As an example, suppose that the conditional probability distribution is not concentrated around a point and has large spread. In other words, the uncertainty is high, and we can find many labels with equally high but numerically small probabilities. In this case, we can choose a small $\beta$ that yields a very high probability  of $ P(Y_{m+1}^{m+n}|X_{m+1}^{m+n}) \leq \beta$. Therefore, the prediction set size scales mainly with with $1/\beta$ and is expected to be large as $\beta$ is small. Intuitively, the left hand side of Theorem 3.1 measures the intrinsic uncertainty and right hand side measures the prediction set size. We have added this discussion to Appendix B after the proof.
>
> * **The goal of our theoretical framework.** We believe that this theoretical framework can be used to evaluate the efficiency of transductive methods in controlled settings. We propose to use our theoretical bounds in synthetic settings, where the conditional probability is known, and the bounds are computable. We can then evaluate how close the existing transductive methods are to the fundamental bound. This is in analogy with research in field of information theory where binary symmetric channels, erasure channels and AWGN channels are used as benchmark for designing error correction codes. We use this vision in our numerical result section, where we evaluated the Bonferroni method and showed its shortcoming. We believe our theory opens up a research program on design and evaluation of transductive prediction methods under different noise regimes.
>
> * **Safety application of transductive methods.** Consider an autonomous driving use case where the goal is to detect humans and animals in the scene. In this scenario, it is highly important that the detector does not miss these objects. A way to improve the performance is to use multiple samples, for instance multiple consecutive camera images or different angles, to detect the objects. In this case, all the samples have the same label, and we can use transductive methods for such predictions.

---

### Official Review · Reviewer_QYpU · 2025-11-01

**Soundness:** 2
**Presentation:** 3
**Contribution:** 2
**Rating:** 4
**Confidence:** 4

**Summary:**

The paper asks how small a joint prediction set can be when doing transductive conformal prediction at a given confidence. It proves lower bounds on expected set size. The main asymptotic message is that to maintain any non-trivial confidence, the expected joint set must grow roughly like exp of n times conditional entropy. A non-asymptotic refinement adds a second order “dispersion” term that depends on the variance of log probabilities. The authors also give an achievability result in an oracle setting where P of Y given X is known, and they analyze a special case where all test points share one true label by connecting to classical hypothesis testing with the generalized Jensen–Shannon divergence. They include a small MNIST toy study to illustrate trends.

**Strengths:**

Puts a clean information-theoretic lens on transductive conformal prediction and tries to sharpen earlier entropy-based limits with a finite sample expansion. The reduction to hypothesis testing in the same-label case is neatly explained and hooks into known optimal tests.

The statements are precise, the asymptotic and finite-blocklength styles are consistent, and the proofs trace to standard tools like information density bounds, Berry–Esseen, and method of types. The paper is careful about what is guaranteed and where the constants come from.

Key quantities like efficiency rate, dispersion, and the role of conditional entropy are defined clearly. The contrast between joint confidence and per-point Bonferroni is made explicit and illustrated. The same-label section is self-contained and readable.

Joint guarantees are relevant in batch certification, safety screening, and ranking. Having even pessimistic lower bounds helps practitioners understand why transductive sets often balloon as n grows. The work could become a common reference when teams debate whether transductive guarantees are worth the price in set size.

**Weaknesses:**

The headline asymptotic bound reaffirms the conditional entropy barrier already highlighted in recent work on conformal efficiency. The finite sample refinement with a dispersion term is welcome, but the experiments show a persistent gap that closes slowly, which makes it hard to see the practical sharpening.

The achievability result assumes oracle access to P of Y given X. That turns the task into thresholding products of true class probabilities and inevitably matches the converse to first and second order. This is informative theoretically but not actionable. The paper stops short of proposing any implementable transductive procedure that approaches the bound when P of Y given X is learned with error.


Efficiency is measured only by expected set size. In transductive practice, teams often optimize other surrogates like false coverage proportion, cost-weighted set size, or rank-based utility. The bounds are said to “extend” to other notions, but those are deferred. Without at least one nontrivial worked out alternative, the generality claim feels thin.

The MNIST label noise toy is not enough. It uses simple noise models where H of Y given X is tractable, then shows that Bonferroni blows up. That result is unsurprising. There is no stress test on real transductive use cases such as ranking or batch classification with covariate shift, no study of how well one can estimate H and dispersion from data, and no attempt to check tightness against a strong transductive algorithm rather than a Bonferroni baseline.

Dispersion is defined as the standard deviation of log P of Y given X. This could be quite useful as a diagnostic, yet the paper does not show how to estimate it reliably from finite data or how it correlates with observed set growth. The reader is left without a recipe to turn the bound into an engineering rule of thumb.

**Questions:**

Can you give a theorem or a corollary that shows your finite sample lower bound strictly dominates prior entropy-only bounds over a clear domain of alpha and n, with explicit constants? A small synthetic where you can compute both exactly would make the gain concrete.

Suppose P of Y given X is approximated by a calibrated classifier with a known risk or Bregman divergence to the truth. Can you translate that misspecification into a slack in the achievability side, even if loose? A bound of the form “gap grows as epsilon to some power” would be valuable.

Pick one alternative efficiency metric, for example expected rank of the true label inside the joint set or a budgeted FCP, and carry your full derivation through to a nontrivial corollary. This would support the claim that the framework covers broader measures.

Can you sketch an extension of the finite sample bound to continuous X with mild regularity on scores, perhaps using empirical process tools instead of types? Even a simplified theorem for plug-in density ratios would widen the impact.

Provide a practical estimator and a confidence band for H of Y given X and dispersion from held-out data, with a study of bias and variance. Then compare the predicted lower bound against observed growth on at least one real task. This would turn the theory into a planning tool.

Compare your bound with the joint set sizes produced by a modern transductive method that is more nuanced than Bonferroni, across a range of n and calibration sizes. Highlight the regimes where the bound is close to achievable and where there is a big gap.

---

> ### Author Response · Authors · 2025-12-03
> **Response to the reviewer - part 1**
>
> We would like to thank the reviewer for the thorough reading of the paper and many fruitful suggestions.
>
> * **Regarding the conditional entropy barrier.** Our result is derived for transductive learning methods. As we mentioned in the paper (line 124), the conditional entropy barrier derived by Correia et al. 2024 is not tight. It only shows that when the log prediction set size is below the conditional entropy, the confidence level cannot be pushed down arbitrarily.  We show a stronger result: it is impossible to get any non-trivial confidence levels. This result cannot be obtained from the analysis of Correia et al. 2024, which is based on Fano's inequality (known to provide weak converse results in information theory).
>
> * **Regarding the second order term.** The dispersion term is essential for practical applications, since we are in non-asymptotic settings. As it can be seen from Figure 1.a, the gap between the asymptotic bound given by the conditional entropy and non-asymptotic bound given by the equation (5) is still considerable, even when we have to predict for 50 test samples. Therefore, practitioners should consider the non-asymptotic bound with dispersion term as guideline.
>
> * **On oracle access to the conditional distribution and an implementable transductive procedure.** It is worth noting that thresholding true class probabilities is known to be optimal in classical, non-transductive, conformal prediction.
>
> Naturally, this has led author to approximate the conditional probability first to construct such sets. A notable example is Sadinle et al., 2019. section 4.1 [1], where the authors use techniques like k Nearest Neighbors, local polynomial estimator and Regularized multinomial logistic regression to approximate the conditional distributions. Such finding applies to transductive setting as well. We have added this discussion in the new version of the paper.
>
> * **Regarding efficiency measures.** We have discussed in Supplementary C that the probability measure $Q$ can be used in general instead of counting measure, which yields the prediction set size as measure of efficiency. We have added another example of alternative measures in the new version of the paper. For instance, S-criterion for efficiency is an example of cost-weighted set size the reviewer mentioned. We added it as an example.
>
> * **Experimental setup.**  We believe that this is the first work that mentions inefficiency of Bonferroni methods. Note that this is not an artifact of the synthetic noise model. Even for very small noise, $\epsilon=0.98$, it can be seen from our figures that the set size blows up even around 10 test samples. We have restricted ourselves to classification use cases. Although ranking is among the first applications of transductive learning, transductive learning can be applied to many other learning problems.
>
> * **Regarding the dominance of the finite sample lower bound.** The difference between non-asymptotic and asymptotic bounds is given precisely in Theorem 3.4: $\log\Delta + \sqrt{n}\sigma\text{Q}^{-1}(\alpha + \rho/\sqrt{n}\sigma^3+ \Delta)$ if $\alpha + \rho/\sqrt{n}\sigma^3+ \Delta\in [0,1]$. Fix any $\alpha\in(0,1/2)$, which is typical for confidence prediction scenarios. As $n$ increases, the term $\rho/\sqrt{n}\sigma^3$ shrinks and we can always pick a $\Delta$ such that $\alpha + \rho/\sqrt{n}\sigma^3+ \Delta\in [0,1/2)$. For this choice, the inverse Q-function is positive, and therefore the overall term  $\log\Delta + \sqrt{n}\sigma\text{Q}^{-1}(\alpha + \rho/\sqrt{n}\sigma^3+ \Delta)$ becomes positive as $n$ increases. This shows that for the confidence level $\alpha < 1/2$, the bound becomes strictly bigger that the conditional entropy. For a concrete and computable example, please see our numerical simulation, Figure 1.a, which shows the the non-asymptotic bound always dominates the conditional entropy (dashed line).
>
> * **Approximate conditional distribution.** We have add a new theorem to the paper based on this suggestion. Interestingly, the dominant additional term in the expected efficiency rate comes from the expected KL-divergence between the ground truth and approximate conditional distribution. Equivalently, if the approximation error in term of this metric is $\epsilon$, then the expected set size will grow with the factor $e^{n\epsilon}$.
>
> * **Regarding alternative efficiency metric.** Let's start by discussing the criterion used in Chapter 3 of Vovk et al. 2022 [2]. Two categories are discussed for efficiency analysis.  One is called observed criteria where the efficiency is measured based on the observation of the data label. The other category of criteria measures efficiency using the prediction set only. Our derivation in Appendix C applied to non-observed criterion of efficiency. We added a few examples to Appendix C for some new metrics. In particular, we have included N-criterion and (modified) S-crieterion based on p-values as two examples.

---

> > ### Author Response · Authors · 2025-12-03
> > **Response to the reviewer - part 2**
> >
> > * **Regarding the extension of method of types.** Many  tools used in method of types, presented in Appendix A., will break as soon as we move to the continuous setting. The key challenge arises wherever we need to use the cardinality of the sample space $\mathcal{X}$. For continuous cases, the cardinality is unbounded. This is an important step in all the proofs. For instance, in appendix E., the proof of Theorem 4.2 has a term $e^{-n\lambda} (n+1)^{|\mathcal{X}|}(N+1)^{|\mathcal{X}|}$ in the upper bound. For infinite sets, the bound does not vanish with $n$. The empirical process theory, for instance bounds like Dvoretzky–Kiefer–Wolfowitz inequality or concentration inequality results for KDE, cannot address this issue, as they only control the approximation error. This is a well known problem in the information theory and large deviation theory.
> >
> > * **Using theory as a planning tool.** This is an excellent insight, and our goal has been similar but with a key difference. We propose to use our theoretical bounds in synthetic settings, where the conditional probability is known, and the bounds are computable. We can then evaluate how close the existing transductive methods are to the fundamental bound. This is in analogy with research in field of information theory where binary symmetric channels, erasure channels and AWGN channels are used as benchmark for designing error correction codes. We use this vision in our numerical result section, where we evaluated the Bonferroni method and showed its shortcoming.
> >
> > We would like to add another comment regarding the approximation of the conditional entropy. Approximating the conditional entropy and dispersion mostly relies on an approximation of the conditional probability distribution. Let's assume that we use a trained model to approximate the conditional probability distribution. If we use the same model both for constructing the prediction sets and approximating the lower bound, the resulting prediction set size will be trivially tight as the lower bound has been computed using the same  (approximate) conditional distribution. One can think of using different models for approximating the entropy and for constructing the prediction sets, however, it is not clear why a good model for approximating entropy cannot be used for building prediction sets.
> >
> > Overall, we believe that it is better to use our theory in synthetic setup to measure the efficiency of transductive methods.
> >
> > * **Regarding the choice of modern transductive methods.** To the best of our knowledge, the existing practical transductive methods rely on Bonferroni principle: compute $p$-values for each test samples independently and then build prediction sets for each test samples (See for instance [3]). As we shown in the paper, these approaches are highly inefficient. We are open to include other methods, if the reviewer has a concrete suggestion in
> >
> > [1] Mauricio Sadinle, Jing Lei, and Larry Wasserman. Least ambiguous set-valued classifiers with
> > bounded error levels. Journal of the American Statistical Association, 114(525):223–234, 2019.
> >
> > [2] Vladimir Vovk, Alexander Gammerman, and Glenn Shafer. Algorithmic Learning in a Random
> > World. Springer International Publishing, 2022.
> >
> > [3] Ulysse Gazin, Gilles Blanchard, and Etienne Roquain. Transductive conformal inference with adaptive scores. In International Conference on Artificial Intelligence and Statistics, pp. 1504–1512.
> > PMLR, 4 2024.

---

### Author Response · Authors · 2025-12-03
**General comment**

We would like to thank all the reviewers again for the time they spent on our paper and their comments and questions.

We have addressed the questions raised by the reviewers and uploaded a new version of the paper with new materials colored in red. We summarize the assessments and concerns of the reviewers and our response below.

* The reviewers acknowledged the precise theoretical analysis of transductive confidence prediction in our information theoretic framework as well as the novelty of the result. We have connected tools from information theory, large deviation theory, hypothesis testing and conformal prediction.

* The reviewers highlighted the importance of the work for practitioners. Most of the questions were related to this aspect of the work: how can we use the theory?

* We have tried to clarify how the theory should be used: similar to information theory tradition, the practical methods can be stress tested in synthetic setups with known noise model and computable entropy and dispersion term. The distance to optimality can then be measured. To start, we showed that Bonferroni methods, the only practical transductive method in the literature to the best of our knowledge, is highly inefficient as the number of samples grow.

* There were some questions about approximating the conditional distribution and its impact on our achievability bounds. We added a new related theorem in the appendix D. We also instantiated our general bound in Appendix C and showed how it can be used to obtain bounds for different efficiency measures. We also added an experimental results related to Section 4.

* The other questions are addressed by clarifying our stance further. We believe these questions do not raise major concern for the paper.

---

### Meta-Review · Area_Chair_MRBE · 2026-01-08

**Summary:**

This paper establishes information-theoretic bounds on the trade-off between confidence and efficiency in transductive conformal prediction. Their bounds show that non-trivial confidence levels imply exponential growth, characterised by conditional entropy and a second-order dispersion term.

The key aspect of the discussion centers on novelty. Reviewer psdB (who has background in information theory) argues that "the results of the analysis appear to be rather expected and limited in scope." and that "The setting studied in Section 4 is a direct extension of Gutman's work". Reviewer QYpU also states "proofs trace to standard tools like information density bounds, Berry–Esseen, and method of types" and they also state "That result is unsurprising". However, Reviewer QYpU seems to have a more favorable view of the contribution. The authors argue that their "theoretical results are a bit more nuanced". Since I am not an expert in information theory, I cannot judge the novelty of the theoretical results and I defer to the opinion of the reviewers. Reviewer MyY1 who gave the highest score is also not an expert in information theory, thus while they appreciate the paper they are not well-positioned to judge the novelty of the theoretical results as well.

A common question was about the approximation error when applying the theoretical results to practical scenarios where the distribution is unknown. The authors addressed this in the revised version by adding Theorem D.2, which provides bounds on the approximation error when using estimated distributions. While this addition strengthens the practical relevance of the theoretical results, I agree with Reviewer QYpU that at the end "The reader is left without a recipe to turn the bound into an engineering rule of thumb".

In general, the authors argue to "use" their bounds in synthetic settings, where the conditional probability is known, and the bounds are computable. However, it's not clear whether this provides additional insight since the exponential blow-up seems to be expected.

In general, while it does seem that results are correct, given that it's unclear how novel and insightful they are beyond existing results, I recommend rejecting the paper. Perhaps, a more theory-centric venue would be more appropriate for this work, or at least the exposition could be improved to make the paper more approachable for the ICLR audience.

**Reviewer Concerns:**

The authors added Theorem D.2 which addressed a common question. They also added: additional examples of alternative efficiency metrics, clarifications on how their approach differs from prior entropy-based bounds. They did not sufficiently address the points regarding: practical guidance for estimation from real data, modern transductive methods beyond Bonferroni (authors claim no better practical methods exist), clearer intuition for non-experts.

**Reviewer Scores:**

Reviewer psdB: Hard to say given that the disagreement is about novelty, but likely they would maintain their score.

Reviewer MyY1: Likely keep their high score.

Reviewer FpE3: Questions only partially addressed, likely keep their score. Only added interpretation for one Theorem but not the others.

Reviewer QYpU: There is some chance that they would have increased their score given that the authors provided a comprehensive reply.

---

### Decision · Program_Chairs · 2026-01-26

Reject